# Transforming Growth Factor-beta signaling in αβ thymocytes promotes negative selection

Mark J. McCarron [1,2,3,4,7,8], Magali Irla [5,8], Arnauld Sergé [6], Saidi M'Homa Soudja[1,2,3,4] & Julien C. Marie[1,2,3,4]*

In the thymus, the T lymphocyte repertoire is purged of a substantial portion of highly self-reactive cells. This negative selection process relies on the strength of TCR-signaling in response to self-peptide-MHC complexes, both in the cortex and medulla regions. However, whether cytokine-signaling contributes to negative selection remains unclear. Here, we report that, in the absence of Transforming Growth Factor beta (TGF-β) signaling in thymocytes, negative selection is significantly impaired. Highly autoreactive thymocytes first escape cortical negative selection and acquire a Th1-like-phenotype. They express high levels of CXCR3, aberrantly accumulate at the cortico-medullary junction and subsequently fail to sustain AIRE expression in the medulla, escaping medullary negative selection. Highly autoreactive thymocytes undergo an atypical maturation program, substantially accumulate in the periphery and induce multiple organ-autoimmune-lesions. Thus, these findings reveal TGF-β in thymocytes as crucial for negative selection with implications for understanding T cell self-tolerance mechanisms.

[1] Department of Immunology Virology and Inflammation, Cancer Research Center of Lyon (CRCL) UMR INSERM1052, CNRS 5286 28 rue Laennec, F-69373 cedex 08 Lyon, France. [2] Université Lyon 1, F-69000 Lyon, France. [3] Centre Léon Bérard, Lyon, F-69008 Lyon, France. [4] Labex DEVweCAN F-69008, Lyon, France. [5] Centre d'Immunologie de Marseille Luminy (CIML), INSERM U1104, CNRS UMR 7280, Aix-Marseille Université UM2, F-13288 cedex 09, Marseille, France. [6] Centre de Recherche en Cancérologie de Marseille, Institut Paoli-Calmettes, INSERM U1068, CNRS UMR7258, Aix-Marseille Université UM105, 13273 cedex 09 Marseille, France. [7] Present address: Genentech, South San Francisco, CA 94080, USA. [8] These authors contributed equally: Mark J. McCarron, Magali Irla. *email: julien.marie@inserm.fr

Autoimmune diseases are typically associated with peripheral αβ T lymphocytes that strongly react against self-antigens expressed by one or several tissues. During their ontogeny in the thymus, αβ T lymphocytes undergo a double selection process: (i) positive selection based on their ability to recognize major histocompatibility complex (MHC) self-peptide complexes and (ii) negative selection, during which a substantial portion of highly autoreactive αβ T lymphocytes are purged[1]. While positive selection is restricted to the cortex, the negative selection process occurs in two waves localized first in the cortex and then in the medulla. In the cortex, recently positively selected autoreactive CD4$^{pos}$CD8$^{pos}$ double-positive (DP) thymocytes are mainly negatively selected on antigens presented by dendritic cells (DCs)[2,3]. Next, cortical thymocytes that passed the first wave of negative selection pursue their maturation program from the DP to the single-positive (SP) stage, where they acquire CCR7 expression allowing their migration from the cortex to the medulla, which is enriched in CCR7 ligands, CCL19 and CCL21[3–6]. There, their physical interaction with medullary thymic epithelial cells (mTECs) leads to mTEC differentiation[7,8]. Mature mTECs have a unique ability to express and present a wide range of tissue-restricted antigens (TRAs) representing almost all peripheral tissue self-antigens on their MHC[9]. This promiscuous gene expression program is regulated, at least in part, by the transcriptional regulator AIRE (AutoImmune REgulator) and largely contributes to the medullary negative selection of highly autoreactive SP thymocytes[10]. The expression of AIRE by mTECs is largely dependent on signaling through members of the Tumor Necrosis Factor Receptor Superfamily (TNFRSF) such as RANK and CD40 in response to their ligands expressed by highly autoreactive thymocytes entering into the medulla[7,8,11].

In both cortical and medullary negative selection, the elimination of highly autoreactive thymocytes is mainly mediated by the pro-apoptotic BH3-only protein (BIM) in response to the strength of signals induced by the TCR-self-peptide-MHC interaction[12]. However, during their ontogeny, thymocytes also integrate other signals, particularly those provided by cytokines. Whether cytokine signaling could contribute to the thymic negative selection remains elusive. Among the cytokines present in the thymus, transforming growth factor beta (TGF-β) is constitutively expressed from the earliest stages of development [13]. This polypeptide is highly conserved in all mammals and signals through a receptor composed of two subunits TGFβ–RI and TGFβ–RII[14]. Upon binding to its receptor, TGF-β induces the kinase activity of TGFβRII, which in turn activates the kinase domain of TGFβ–RI. TGF-β signaling involves either the canonical pathways, which are SMAD4 and/or TRIM33-dependent, or a SMAD4–TRIM33 independent pathway involving mainly the MAPK/ERK cascade[15,16]. There are three forms of TGF-β (1–3), and all share the same receptor complex. However, within the immune system, including the thymus, TGF-β1 is prevalent[14]. Strikingly, TGF-β1 is the only cytokine whose deprivation leads to fatal autoimmune disorders[17]. We and others revealed that *CD4-cre;tgfbr2$^{fl/fl}$* mice, which lack TGF-β signaling in αβ T lymphocytes from the DP stage in the thymus, develop severe early-onset multi-organ autoimmunity, including massive production of auto-antibodies, that cannot be rescued by the presence of wild-type Foxp3$^+$ regulatory T (Treg) cells[18–20]. Interestingly, the ablation of TGF-β signaling later after negative selection, including in peripheral mature T lymphocytes, failed to induce autoimmunity[21,22], arguing against a direct role for TGF-β signaling in the control of peripheral T-cell tolerance and suggesting a potential role of this cytokine in the negative selection process.

In order to determine the role of TGF-β signaling in negative selection, we carefully analyzed the development steps of thymocytes in *CD4-cre;tgfbr2$^{fl/fl}$* mice. Here, we reveal a critical role for TGF-β signaling in αβ thymocytes to promote their negative selection. As a consequence of TGF-β signaling deprivation, and particularly of its SMAD4/TRIM33-independent pathway, a substantial proportion of highly autoreactive thymocytes escape BIM-dependent apoptosis during the cortical negative selection process. They express the transcription factor T-Bet and the chemokine receptor CXCR3, aberrantly accumulate at the cortico-medullary junction (CMJ), fail to induce AIRE$^{pos}$ mTEC differentiation and TRA expression, and thus escape medullary negative selection. Non-eliminated highly autoreactive thymocytes rapidly undergo maturation before accumulating in the periphery causing multi-organ autoimmune lesions. Overall, our results unravel a fundamental mechanism of central self-tolerance, highlighting TGF-β as the first described cytokine essential for thymic negative selection.

## Results

**TGF-β signaling in thymocytes promotes cortical and medullary negative selection.** In order to interrogate the role of TGF-β signaling in the two waves of negative selection of αβ thymocytes, we used *Foxp3$^{gfp}$;CD4-cre;tgfbr2$^{fl/fl}$* (TGFβR-KO) mice in which TGF-β signaling was selectively abrogated in αβ thymocytes at the early DP stage, just prior to negative selection[23,19]. Both CD4$^{pos}$ CD8$^{low}$ thymocytes and CD4$^{pos}$ CD8$^{neg}$ (CD4 SP) thymocytes, which, based on their CCR7 expression, correspond to cortical and medullary thymocytes, respectively[4,5], were analyzed (Fig. 1a, b). We took advantage of a previous observation that, in an unskewed repertoire, CD4$^{pos}$ highly autoreactive non-regulatory thymocytes have been defined by the expression of the Ikaros family transcription factor, Helios, and the absence of Foxp3 expression[3]. Foxp3$^{neg}$ Helios$^{pos}$ thymocytes were reported to undergo both cortical and medullary negative selection[3]. Interestingly, we observed a three- to five-fold increase, in the percentage and absolute numbers of Foxp3$^{neg}$ Helios$^{pos}$ CD4$^{pos}$ CD8$^{low}$ thymocytes in TGFβR-KO mice compared to TGFβR-WT mice, whereas the presence of Foxp3$^{neg}$ Helios$^{pos}$ CD4 SP cells was halved in TGFβR-KO mice (Fig. 1c, d). The alteration of Foxp3$^{neg}$ Helios$^{pos}$ CD4$^{pos}$ CD8$^{low}$ cell homeostasis in TGFβR-KO mice was not due to potential side effects from the systemic inflammation that can occur in TGFβR-KO neonates, or due to exacerbated re-circulation of activated peripheral lymphocytes into the thymus, since fetal thymic organ culture (FTOC) also led to the over-representation of Foxp3$^{neg}$ Helios$^{pos}$ CD4$^{pos}$ CD8$^{low}$ thymocytes in TGFβR-KO mice (Supplementary Fig. 1). Moreover, we failed to explain the aberrant proportion of Foxp3$^{neg}$ Helios$^{pos}$ CD4$^{pos}$ CD8$^{low}$ thymocytes by a higher proliferative rate based on Ki67 staining, or by exaggerated positive selection, analyzed by the expression levels of CD69, CD5 and TCR on DP thymocytes (Supplementary Fig. 2a, b).

Given the key role of BIM in the negative selection of Foxp3$^{neg}$ Helios$^{pos}$ thymocytes[3], we next assessed the effects of TGF-β signaling on the expression of this pro-apoptotic factor. In clear contrast to TGFβR-WT mice, the portion of BIM$^{pos}$ cells among the Foxp3$^{neg}$ Helios$^{pos}$ CD4$^{pos}$ CD8$^{low}$ cells was 3–4 times lower in TGFβR-KO mice (Fig. 1e, f). Consistent with this observation, caspase 3 activation in Foxp3$^{neg}$ Helios$^{pos}$ CD4$^{low}$ CD8$^{low}$ thymocytes was also decreased in TGFβR-KO mice compared to TGFβR-WT mice (Fig. 1f). Though Foxp3$^{neg}$ Helios$^{pos}$ CD4 SP cells were under-represented in TGFβR-KO mice, their ability to express BIM and undergo apoptosis was also impaired compared to TGFβR-WT mice (Fig. 1e, f). Notably, similar expression of BIM was observed in Foxp3$^{neg}$ Helios$^{neg}$ thymocytes of TGFβR-KO mice and TGFβR-WT mice (Fig. 1e), implying that TGF-β signaling selectively affects the survival of highly autoreactive thymocytes.

In order to determine which branch of TGF-β signaling promotes the negative selection of Foxp3$^{neg}$ Helios$^{pos}$

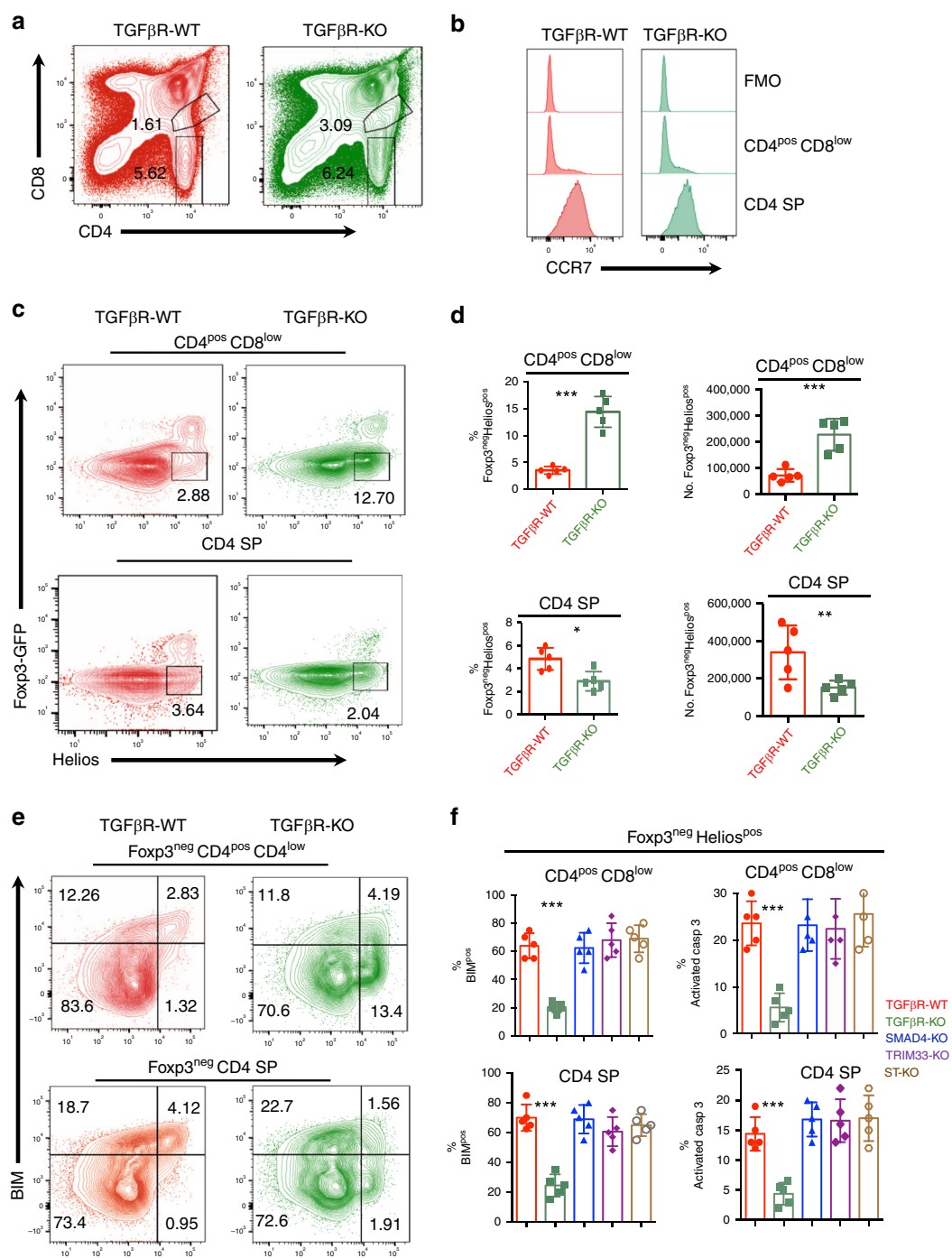

**Fig. 1 TGF-β signaling in thymocytes promotes cortical and medullary negative selection.** Flow cytometry analysis was performed on thymus from TGFβR-KO mice, SMAD4-KO mice, TRIM33-KO mice, ST-KO mice and their wild-type littermate controls. **a**, **b** Gating strategy for flow cytometry analysis of CD4$^{pos}$ CD8$^{low}$ and CD4 SP thymocytes. **c** Representative contour plot analysis of the expression of Foxp3 and Helios among the CD4$^{pos}$ CD8$^{low}$ and CD4 SP thymocytes. **d** Graphs illustrate the proportion of Foxp3$^{neg}$ Helios$^{pos}$ among CD4$^{pos}$ CD8$^{low}$ and CD4 SP thymocytes as well as their absolute numbers. **e** Representative contour plot analysis of the expression of BIM and Helios in Foxp3$^{neg}$ CD4$^{pos}$ CD8$^{low}$ thymocytes and in Foxp3$^{neg}$ CD4 SP thymocytes. **f** Graphs illustrate the percentages of cells expressing BIM and positive for activated caspase 3 among Foxp3$^{neg}$ Helios CD4$^{pos}$ CD8$^{low}$ thymocytes and in Foxp3$^{neg}$ Helios$^{pos}$ CD4 SP thymocytes. All the experiments were conducted in 7–10-day-old animals. Data are representative of three experiments with 3–5 mice per group. *$P < 0.05$, **$P < 0.01$, ***$P < 0.001$ (two-tailed Student's $t$ test). Error bar, mean ± SEM. Source data are provided as a Source Data file.

thymocytes, we next monitored both BIM expression and caspase 3 activation in *Foxp3$^{gfp}$;CD4-cre;smad4$^{fl/fl}$* (SMAD4-KO) mice, *Foxp3$^{gfp}$;CD4-cre;trim33$^{fl/fl}$* (TRIM33-KO) mice, and *Foxp3$^{gfp}$; CD4-cre;smad4$^{fl/fl}$; trim33$^{fl/fl}$* (ST-KO) mice (Fig. 1f). Neither the deprivation of SMAD4, nor TRIM33, nor the deprivation of both

impaired BIM expression and caspase 3 activation in Foxp3$^{neg}$ Helios$^{pos}$ thymocytes, incriminating the SMAD4/TRIM33-independent pathway of TGF-β signaling in the control of thymic negative selection. Together, these observations reveal that TGF-β signaling in thymocytes, through its SMAD4/TRIM33-

independent signaling branch, contributes to both waves of negative selection of Foxp3$^{neg}$ Helios$^{pos}$ thymocytes.

**TGF-β signaling in thymocytes promotes AIRE$^{pos}$ mTEC differentiation.** In order to analyze the cell-autonomous mechanisms responsible for the defect of negative selection in TGFβR-KO mice, we next performed a TGFβR-WT:TGFβR-KO (50:50) mixed bone marrow (BM) chimera. As previously described, these BM chimeric mice developed autoimmunity and inflammation similarly to TGF-βR-KO mice[18,19]. Remarkably, BIM expression was not altered in Foxp3$^{neg}$ Helios$^{pos}$ thymocytes derived from TGFβR-WT BM. This rules out again the possibility that disruption of Foxp3$^{neg}$ Helios$^{pos}$ TGFβR-KO thymocyte homeostasis was a side effect of the inflammation that occurs in TGFβR-KO mice. Interestingly, the presence of TGF-βR-WT cells was sufficient to restore BIM expression in a large fraction of Foxp3$^{neg}$ Helios$^{pos}$ CD4 SP TGFβR-KO thymocytes but not in Foxp3$^{neg}$ Helios$^{pos}$ CD4$^{pos}$ CD8$^{low}$ TGFβR-KO thymocytes (Fig. 2a). Thus, these data strongly suggest that in contrast to cortical negative selection, the defect in medullary selection in TGFβR-KO mice could be due to an indirect effect and likely due to the inability of Foxp3$^{neg}$ Helios$^{pos}$ thymocytes to make the medulla microenvironment suitable for efficient negative selection.

To test this assumption, we next examined the thymic medulla of TGFβR-KO mice. We failed to find any difference either in the size of the medullas or in mTEC cellularity between TGFβR-WT mice and TGF-βR-KO mice (Supplementary Fig. 3a–d). However, strikingly, the density of AIRE$^{pos}$ mTEC, their frequency and absolute numbers were four- to five-fold reduced in the thymus of TGFβR-KO mice compared to TGFβR-WT mice (Fig. 2b, c). Moreover, both the percentage and number of AIRE$^{pos}$ mTEC expressing the maturation associated marker CD80 (ref. [24]) were 3–4 times lower in TGFβR-KO mice compared to TGFβR-WT mice (Fig. 2d). In line with the profound defect in AIRE$^{pos}$ mTEC differentiation in TGFβR-KO mice, the expression levels of genes encoding for TRAs specifically regulated by AIRE were impaired in mTEC from TGFβR-KO mice compared to TGFβR-WT mice (Fig. 2e). In addition to re-establishing the negative selection of Foxp3$^{neg}$ Helios$^{pos}$ CD4 SP TGFβR-KO thymocytes in TGFβR-WT:TGFβR-KO BM chimeric mice (Fig. 2a), the presence of TGFβR-WT thymocytes also rescued the defect in AIRE$^{pos}$ mTEC observed when only TGFβR-KO thymocytes were present (Fig. 2f), ruling out a role for inflammation on the defective mTEC differentiation. Notably, we also found that in AIRE-KO mice only the negative selection of Foxp3$^{neg}$ Helios$^{pos}$ CD4 SP was affected, which is consistent with their CCR7 expression, known to drive their medullary localization (Figs. 2g and 1b)[3–5] implying that in TGFβR-KO mice, the defect in AIRE$^{pos}$ mTEC did not contribute to the impairment of the cortical negative selection. Altogether, these data reveal that in addition to contributing to cortical negative selection, TGF-β signaling allows thymocytes to promote AIRE$^{pos}$ mTEC differentiation and thus to potentiate the expression of TRAs in the medulla that are essential to purge the T-cell repertoire of highly autoreactive cells.

**TGF-β prevents aberrant positioning of highly autoreactive thymocytes.** To better understand the mechanisms by which AIRE$^{pos}$ mTEC differentiation was controlled by TGF-β signaling in thymocytes, we next analyzed the localization of Foxp3$^{neg}$ Helios$^{pos}$ thymocytes. Strikingly, immunohistology analysis revealed that contrary to TGFβR-WT mice, a large fraction of Foxp3$^{neg}$ Helios$^{pos}$ thymocytes were settled at the CMJ in TGFβR-KO mice (Fig. 3a, b), a thymic region described to be enriched in CXCR3-ligands such as CXCL10 (ref. [25]). In agreement with this aberrant localization, we found that in the absence of TGF-β

signaling, around 60% of Foxp3$^{neg}$ Helios$^{pos}$ CD4$^{pos}$ CD8$^{low}$ thymocytes and 40% of Foxp3$^{neg}$ Helios$^{pos}$ CD4 SP thymocytes expressed CXCR3 at their surface, whereas the expression of this chemokine receptor was barely detectable on Foxp3$^{neg}$ Helios$^{pos}$ thymocytes from TGFβR-WT mice (Fig. 3c, d). Interestingly, while BIM was expressed independently of the presence of CXCR3 at the surface of Foxp3$^{neg}$ Helios$^{pos}$ CD4$^{pos}$ CD8$^{low}$ thymocytes, its expression was mainly restricted to the CXCR3$^{neg}$ fraction of Foxp3$^{neg}$ Helios$^{pos}$ CD4 SP thymocytes, implying that Foxp3$^{neg}$ Helios$^{pos}$ CD4 SP that settled at the CMJ in TGFβR-KO mice escaped medullary negative selection (Fig. 3c, d). Importantly, among the Foxp3$^{neg}$ Helios$^{pos}$ CD4 SP thymocytes from TGFβR-WT:TGFβR-KO mixed BM chimera mice, the overexpression of CXCR3 was clearly restricted to TGFβR-KO thymocytes, confirming the intrinsic role of TGF-β signaling in the repression of CXCR3 (Supplementary Fig. 4).

In order to investigate the molecular mechanisms by which TGF-β signaling in thymocytes controls the expression of CXCR3 at the surface of Foxp3$^{neg}$ Helios$^{pos}$ cells, we then monitored the expression of T-Bet, a transcription factor that controls *cxcr3* expression[26]. We found that in TGFβR-KO mice, T-Bet was expressed in both Foxp3$^{neg}$ Helios$^{pos}$ CD4$^{pos}$ CD8$^{low}$ thymocytes and Foxp3$^{neg}$ Helios$^{pos}$ CD4 SP thymocyte, suggesting an early polarization toward a "Th1-like" phenotype of highly autoreactive thymocytes (Fig. 3e, f). Moreover, our data reveal that TGF-β signaling repressed T-Bet expression via its non-canonical branch, since T-Bet expression levels in SMAD4-KO mice, TRIM33-KO mice, and ST-KO mice were similar to those of observed in their littermate controls (Fig. 3f). Overall, these series of experiments point out that in the absence of TGF-β signaling, a large fraction of Foxp3$^{neg}$ Helios$^{pos}$ thymocytes are early polarized toward a "Th1-like" phenotype and are aberrantly positioned at the CMJ, impairing physical interactions with mTEC and escaping to medullary negative selection.

**TGF-β promotes RANK-L expression in highly autoreactive thymocytes.** The histology analysis of the thymus demonstrated that a portion of Foxp3$^{neg}$Helios$^{pos}$ thymocytes still entered into the medulla of TGFβR-KO mice (Fig. 3a). This observation was in line with the presence of CXCR3$^{neg}$ (40–60%) among Foxp3$^{neg}$Helios$^{pos}$ TGFβR-KO thymocytes (Fig. 3c, d), which likely were not trapped at the CMJ, and suggested that the fraction of Foxp3$^{neg}$Helios$^{pos}$ TGFβR-KO thymocytes entering the medulla could be unable to sustain a normal expression of AIRE in mTECs, which relies on the activation of non-canonical NF-kB pathway upon RANK and/or CD40 signaling[24]. We next assessed whether TGF-β signaling in thymocytes affected the expression of these TNFSR ligands. We failed to find any difference in CD40L expression between Foxp3$^{neg}$Helios$^{pos}$ thymocytes from TGFβR-WT mice and TGFβR-KO mice (Supplementary Fig. 5). However, the production of RANKL, which acts in synergy with CD40L to induce AIRE expression in mTEC[7,11], was three to four times selectively decreased in highly autoreactive thymocytes, a population we previously reported to be the key thymocytes providing signals for mTEC differentiation[7] (Fig. 4a, b). Furthermore, the absence of either SMAD4, TRIM33 or both in thymocytes failed to affect RANKL expression in both Foxp3$^{neg}$ Helios$^{pos}$ CD4$^{pos}$ CD8$^{low}$ thymocytes and Foxp3$^{neg}$ Helios$^{pos}$CD4 SP (Fig. 4b) suggesting a role of TGF-β signaling in the control of RANKL expression.

In order to understand how TGF-β signaling promotes RANKL expression in Foxp3$^{neg}$ Helios$^{pos}$ thymocytes, we took advantage of the observation that in Foxp3$^{neg}$ CD4$^{pos}$ CD8$^{low}$ thymocytes the surface marker PD1 was concomitantly expressed with Helios[3] (Fig. 4c). Purified Foxp3$^{neg}$ PD1$^{pos}$ CD4$^{pos}$ CD8$^{low}$

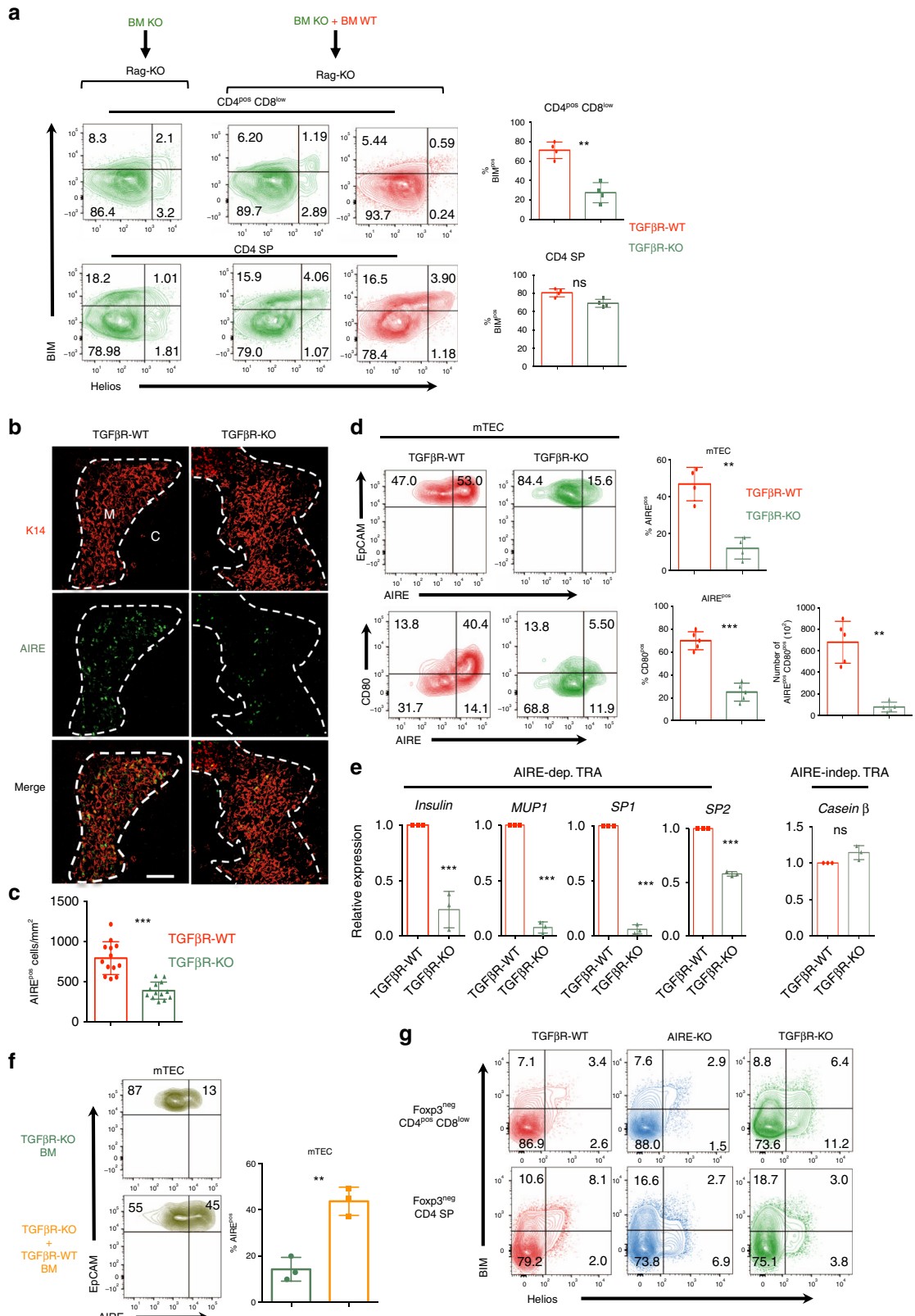

TGFβR-WT thymocytes were cultured in the presence of recombinant TGF-β and inhibitors of TGF-β signaling. We found that TGF-β signaling was sufficient to increase *Rankl* expression in Foxp3neg PD-1pos CD4pos CD8low TGFβR-WT thymocytes (Fig. 4d). Moreover, the upregulation of *Rankl* expression by TGF-β was totally prevented in the presence of a

MAPK pathway inhibitor. (Fig. 4d). Thus, TGF-β signaling, via its SMAD4/TRIM33-independent pathway in thymocytes, not only allows Foxp3neg Heliospos cells to physically interact with mTEC, but also endows them with high expression of RANKL, two essential features known to contribute to a full AIREpos mTEC differentiation.

**Fig. 2 Defective AIRE$^+$ mTEC differentiation in TGF-βR-KO mice. a** T-cell depleted BM cells from TGF-βR-KO mice were mixed or not with T-cell-depleted BM cells from congenic (CD45.1) TGF-βR-WT at a 1:1 ratio ($10^6$ cells each) and transferred into irradiated Rag2-KO mice. About 4.5–5 weeks later, thymi were analyzed by flow cytometry based on gating strategy shown in Supplementary Fig. 4 and graphs demonstrate the percentage of BIM$^{pos}$ cells among the CD4$^{pos}$ CD8$^{low}$ cells and CD4 SP cells. **b, c** Immunostaining on thymus sections from TGF-βR-KO mice and their littermate controls (TGF-βR-WT). The dashed line delineates the medulla as defined by K14 staining (red). C and M denote the cortex and the medulla, respectively. Scale bars, 200 μm). **c** The graph illustrates the density of AIRE$^+$ mTEC cells within the medulla. **d** Flow cytometry analysis of mTECs (CD45$^{neg}$ Ly51$^{low}$ EpCAM$^{pos}$ cells) from TGF-βR-KO mice and TGF-βR-WT mice from gating strategy illustrated in Supplementary Fig. 3c. Graphs demonstrate the percentage of AIRE$^{pos}$ mTECs, the percentage of CD80$^{pos}$ among AIRE$^{pos}$ mTECs and absolute numbers of CD80$^{pos}$ AIRE$^{pos}$ mTECs. **e** Graphs illustrate the relative expression levels of TRA genes in FACS-sorted mTECs after qPCR analysis and normalization with actin expression (value 1 was attributed to the gene expression of mTECs from TGF-βR-WT mice). Dots represent a pool of purified mTECs from 5 mice from gating strategy illustrated in Supplementary Fig. 3c. **f** BM chimeric mice were generated as in **a**. Contour plots represent AIRE expression in mTECs and graph demonstrates the percentage of AIRE$^{pos}$ mTECs. **g** Flow cytometry analysis of Foxp3$^{neg}$ CD4$^{pos}$ CD8$^{low}$ thymocytes and Foxp3$^{neg}$ CD4 SP thymocytes for BIM and Helios expression. Except for chimeric mice, all the experiments were conducted in 7–10-day-old animals. Data are representative of 3–5 experiments, with 3–5 mice per group. ***$P < 0.001$ (two-tailed Student's $t$ test). Error bar, mean ± SEM. Ns: statistically not significant. Source data are provided as a Source Data file.

**TGF-β prevents maturation and thymus egress of highly autoreactive T cells.** In parallel to their selection processes, thymocytes undergo maturation, which is demonstrable by sequential changes in surface molecule expression. The combination of CD69 and MHC-I has been proposed to identify semi-mature thymocytes (SM) (CD69$^+$ MHC-I$^{low}$), mature M1 thymocytes (CD69$^+$ MHC-I$^{high}$) and more mature M2 thymocytes (CD69$^-$ MHC-I$^{high}$)[27]. Thus, we next addressed whether Foxp3$^{neg}$ Helios$^{pos}$ CD4 SP TGFβR-KO thymocytes, which escaped negative selection, completed their maturation process. Strikingly, we observed a three- to four-fold increase in M2 cells among Foxp3$^{neg}$ Helios$^{pos}$ CD4 SP thymocytes from TGFβR-KO mice compared to TGFβR-WT animals. Moreover, in line with an exacerbated maturation, Foxp3$^{neg}$ Helios$^{pos}$ CD4 SP at the SM stage were barely detectable in TGFβR-KO mice (Fig. 5a). Importantly, this exacerbated maturation was restricted to the Foxp3$^{neg}$ Helios$^{pos}$ compartment, since the proportions of M2, M1 and SM cells among Foxp3$^{neg}$ Helios$^{neg}$ CD4 SP thymocytes were similar between TGFβR-KO and TGFβR-WT mice. In addition to the ability of substantial numbers of Foxp3$^{neg}$ Helios$^{pos}$ CD4 SP to reach the M2 stage, around 70–90% of them expressed high levels of Qa2 including the CD69$^{pos}$ fraction (Fig. 5b). Notably, Foxp3$^{neg}$ Helios$^{pos}$ CD4 SP thymocytes were more enriched in mature cells than Foxp3$^{neg}$ Helios$^{neg}$ CD4 SP in both TGFβR-KO mice and TGFβR-WT mice, a feature shared by thymic Treg cells[28], suggesting that highly autoreactive thymocytes that escape negative selection could reach more mature stages than their WT counterparts. In line with this idea, we found that the mature Foxp3$^{neg}$ Helios$^{pos}$ CD4 SP thymocytes (CD69$^{neg}$ or HSA$^{low}$) composed the major part of BIM$^{neg}$ cells (Fig. 5c). Moreover, in agreement with Fig. 3c, d, CXCR3 expression was largely restricted to mature Foxp3$^{neg}$Helios$^{pos}$ CD4 SP TGFβR-KO thymocytes, which were settled at the CMJ and escaped medullary negative selection (Fig. 5b). However, it was notable that in TGFβR-KO mice, the majority of CD69$^{neg}$ Foxp3$^{neg}$ Helios$^{pos}$ CD4 SP thymocytes were CXCR3$^{neg}$ (Fig. 5b), implying that mature CD4 SP were not retained at the CMJ and thus could leave the thymus. In agreement with this assumption, and their M2 phenotype, around 50% of Foxp3$^{neg}$ Helios$^{pos}$ CD4 SP thymocytes from TGFβR-KO mice upregulated the sphingosine-1-phosphate receptor (S1P1) (Fig. 5e), essential for mature thymocyte egress and whose surface expression is repressed by CD69[29,30]. In order to confirm the ability of Foxp3$^{neg}$ Helios$^{pos}$ CD4 SP thymocytes from TGFβR-KO mice to exit the thymus, we performed FITC intra-thymic injections that are classically used to follow emigrating thymocytes[31]. In line with their S1P1 expression, we found that Foxp3$^{neg}$ Helios$^{pos}$ CD4 SP TGFβR-KO thymocytes massively reached the periphery compared to Foxp3$^{neg}$ Helios$^{pos}$ CD4 SP TGFβR-WT

thymocytes, whereas global thymus egress seemed to be unaffected (Fig. 5f). This massive thymus egress likely explains the defect of Foxp3$^{neg}$ Helios$^{pos}$ CD4 SP thymocytes in TGFβR-KO mice observed in Fig. 1c, d.

Finally, given that the ablation of TGFβR in CD4$^{pos}$ peripheral T lymphocytes, which had undergone negative selection, was reported to fail to induce autoimmunity[21], we assessed whether mature CD4 SP TGFβR-KO thymocytes, which were not purged of Foxp3$^{neg}$ Helios$^{pos}$ thymocytes, can induce autoimmune disorders. To this end, we reconstituted the T-cell compartment of CD3-KO mice with mature CD4$^{pos}$ thymocytes derived from either TGFβR-KO or TGFβR-WT mice (Fig. 6). Remarkably, 8 weeks after cell transfer, in clear contrast with mice transferred with TGFβR-WT thymocytes, all the recipient mice that received TGFβR-KO thymocytes presented massive tissue infiltrations. Classically, these infiltrations were observed in the Langerhans islets with insulitis without cell-destruction, corresponding to pre-diabetes, the derma and the colonic lamina propria of all animal recipients reconstituted with TGFβR-KO thymocytes (Fig. 6a Table 1) and were associated with IFN-γ and TNFα production by CD4$^+$ T lymphocytes infiltrating the tissues (Fig. 6b). Thus, in the absence of TGF-β signaling, a large portion of highly autoreactive thymocytes are not eliminated, progress to mature stages, leave the thymus and induce multiple organ autoimmunity.

## Discussion

In mammals, immune tolerance to self-antigens is primarily induced in the thymus through negative selection. This key process has long been thought to be only mediated by the strength of TCR signaling in response to self-antigens presented in the thymus[32]. Our findings demonstrate that negative selection, both in the cortex and in the medulla, is also dependent on TGF-β signaling in thymocytes, revealing an unsuspected role of this cytokine in this pivotal event for central tolerance (Fig. 7).

The impairment of cortical negative selection in the TGFβR-KO mice strongly suggests that the TGF-β-dependent program leading to the elimination of highly autoreactive thymocytes is likely acquired during, or shortly after, their positive selection in the cortex. However, TGF-β signaling should be maintained through thymocyte development, including in the medulla since it is essential for the elimination of highly autoreactive thymocytes there. Notably, in the thymus, TGF-β is secreted by a large panel of cells in its inactive form composed of TGF-β linked to the latent associated protein (LAP), which covers all contact sites of TGFβ that must interact with TGFβR to induce signaling. LAP is enriched in RGD sequences and the activation of TGF-β1 form is clearly associated with integrin αvβ8 (ref. [33]). However, the identification of the cells allowing TGF-β1 activation both in the

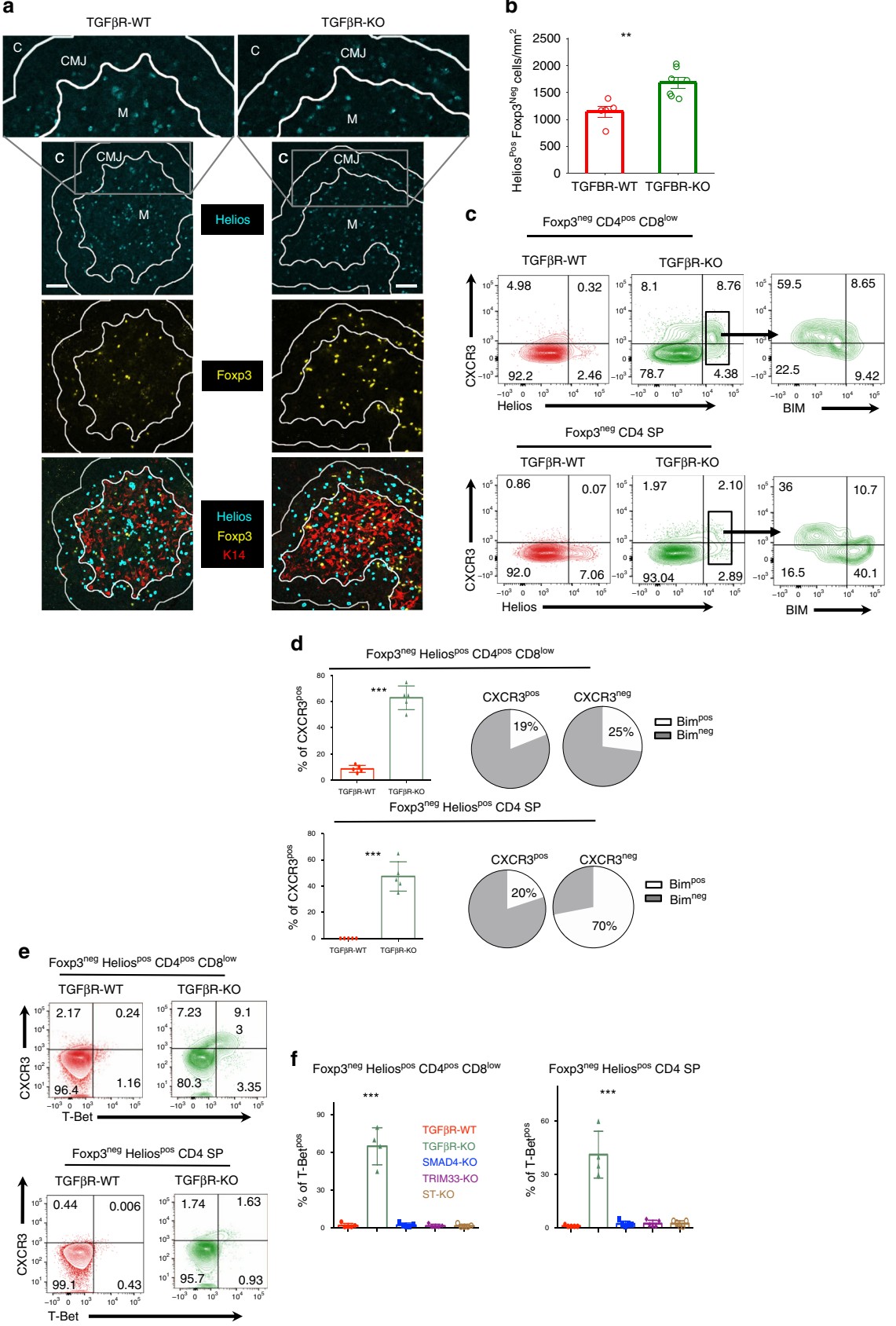

**Fig. 3 Aberrant positioning of Helios$^{pos}$ thymocytes in the absence of TGF-β signaling. a** Immunostaining on thymic sections from TGFβR-KO mice and their littermate controls (TGFβR-WT). The medulla (M) was stained with K14 (red). The cortex and the cortico-medullary junction are noted C and CMJ, respectively. The CMJ was defined using computational approaches. Foxp3$^{pos}$ cells are stained in yellow and Helios$^{pos}$ cells in blue. Scale bars, 50 μm (magnication 3x). **b** Graph shows the density of Foxp3$^{neg}$ Helios$^{pos}$ cells at the CMJ. **c, d** Flow cytometry analysis on Foxp3$^{neg}$ CD4$^{pos}$ CD8$^{low}$ thymocytes and Foxp3$^{neg}$ CD4 SP thymocytes from TGF-βR-KO mice and TGF-βR-WT mice based on gating strategy of Fig. 1. **c** Countor plots illustrate the expression of CXCR3 and Helios, whereas contour plots represent the expression of BIM and CXCR3 among either Foxp3$^{neg}$ Helios$^{pos}$ CD4$^{pos}$ CD8$^{low}$ thymocytes or Foxp3$^{neg}$ Helios$^{pos}$ CD4 SP thymocytes. **d** Histograms illustrate the percentage of CXCR3$^{pos}$ cells among either Foxp3$^{neg}$ Helios$^{pos}$ CD4$^{pos}$ CD8$^{low}$ thymocytes or Foxp3$^{neg}$ Helios$^{pos}$ CD4 SP thymocytes. Pie graphs demonstrate the expression of BIM relevant to CXCR3 expression. **e, f** Representative flow cytometry analysis of CXCR3 and T-Bet expression among either Foxp3$^{neg}$ Helios$^{pos}$ CD4$^{pos}$ CD8$^{low}$ thymocytes or Foxp3$^{neg}$ Helios$^{pos}$ CD4 SP thymocytes. Histograms illustrate the percentage of T-Bet$^{pos}$ cells and CXCR3$^{pos}$ cells among Foxp3$^{neg}$ Helios$^{pos}$ CD4$^{pos}$ CD8$^{low}$ thymocytes and Foxp3$^{neg}$ Helios$^{pos}$ CD4 SP thymocytes in *CD4-cre;smad4$^{fl/fl}$* (SMAD4-KO) mice, *CD4-cre;trim33$^{fl/fl}$* (TRIM33-KO) mice, *CD4-cre;smad4$^{fl/fl}$; trim33$^{fl/fl}$* (ST-KO) mice. All the experiments were conducted in 7–10-day-old animals. Data are representative of 3 experiments, with 3–4 mice per group. **P < 0.01, ***P < 0.001 (two-tailed Student's *t* test). Error bar, mean ± SEM. Source data are provided as a Source Data file.

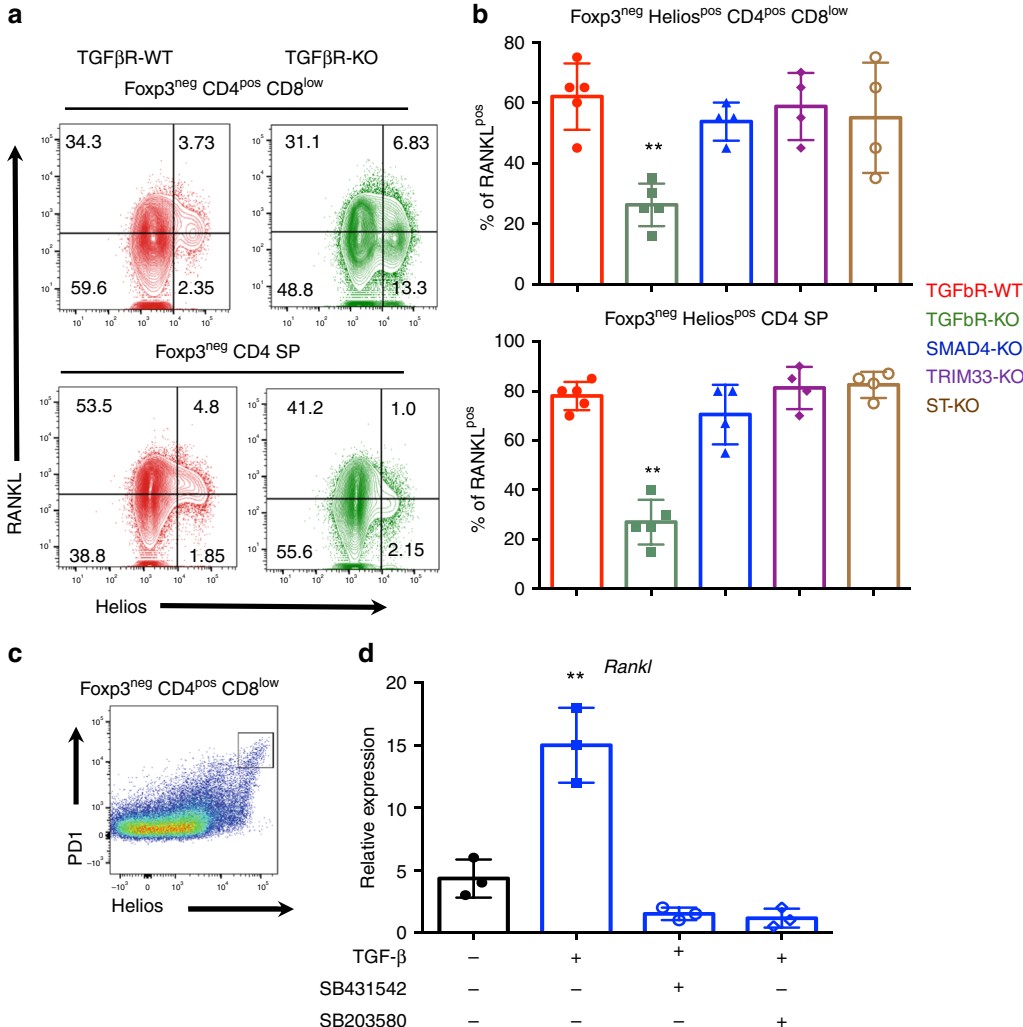

**Fig. 4 TGF-β induces RANK-L expression in Helios$^{pos}$ thymocytes. a**, **b** Flow cytometry analysis of RANK-L and Helios expression in Foxp3$^{neg}$ CD4$^{pos}$ CD8$^{low}$ thymocytes and Foxp3$^{neg}$ CD4 SP thymocytes based on gating strategy depicted in Fig. 1a. **b** Foxp3$^{neg}$ CD4$^{pos}$ CD8$^{low}$ thymocytes and Foxp3$^{neg}$ CD4 SP thymocytes from CD4-cre;smad4$^{fl/fl}$ (SMAD4-KO) mice, CD4-cre;trim33$^{fl/fl}$ (TRIM33-KO) mice, CD4-cre;smad4$^{fl/fl}$; trim33$^{fl/fl}$ (ST-KO) mice were analyzed by flow cytometry for RANKL expression. Histograms demonstrate the percentages of RANKL$^{pos}$ among the two populations. **c** FACS sort gating strategy. **d** FACS-sorted Foxp3$^{neg}$ PD1$^{pos}$ CD4$^{pos}$ CD8$^{low}$ thymocytes from TGFβR-WT mice were then cultured for 3–4 h with anti-CD3 antibodies to mimic TCR stimulation in the presence of TGF-β, TGFβ-R inhibitor (SB431542) and p38 MAPK inhibitor (SB203580). Histograms illustrate the relative expression of *rank-l* measured by q-RT-PCR and normalized to *gadph* expression. All the experiments were conducted in 7–10-day-old animals. Data are representative of 2–3 experiments, with 3–5 mice per group. *P < 0.05, **P < 0.01, ***P < 0.001 (two-tailed Student's *t* test). Error bar, mean ± SEM. Source data are provided as a Source Data file.

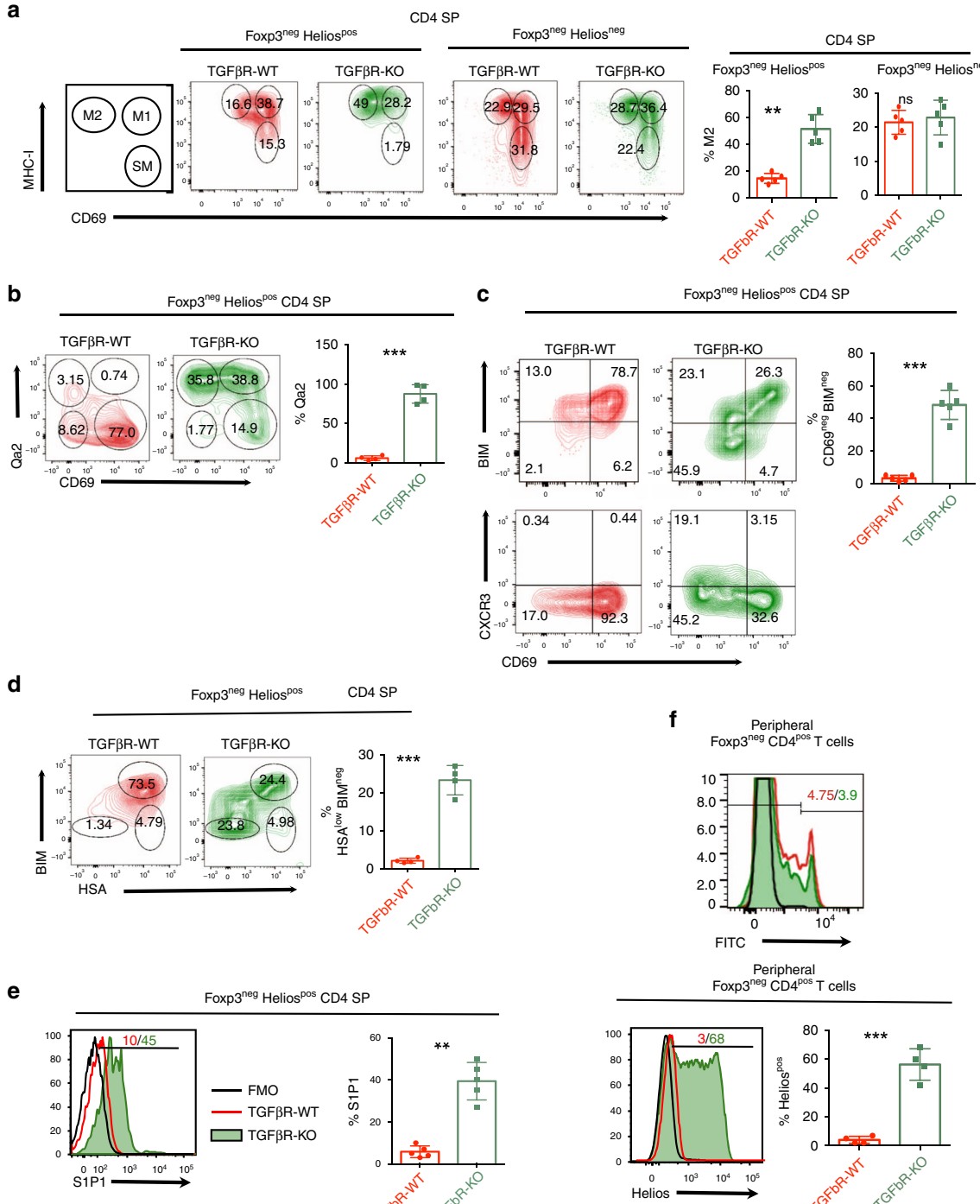

**Fig. 5 TGF-β prevents exacerbated maturation and thymic egress of Helios[pos] thymocytes. a–e** Flow cytometry analysis based on gating strategy depicted in Fig. 1a, b. **a** Flow cytometry contour plots illustrating the SM-M1-M2 populations among Foxp3[neg] Helios[pos] CD4 SP thymocytes and Foxp3[neg] Helios[neg] CD4 SP from TGF-βR-KO mice and TGF-βR-WT mice. Histograms demonstrate percentage of M2 cells. **b** Flow cytometry analysis of Qa-2 and CD69 expression at the surface of Foxp3[neg] Helios[pos] CD4 SP thymocytes from TGF-βR-KO mice and TGF-βR-WT mice. **b** Histogram illustrates percentage of Qa2[pos] cells. **c** Flow cytometry contour plots demonstrating the expression of BIM, CD69 and CXCR3 in Foxp3[neg] Helios[pos] CD4 SP thymocytes from TGF-βR-KO mice and TGF-βR-WT mice. Histogram illustrates the percentage of CD69[neg] BIM[pos] cells among Foxp3[neg] Helios[pos] CD4 SP thymocytes. **d** Flow cytometry analysis of HSA (CD24) and BIM expression on Foxp3[neg] Helios[pos] CD4 SP thymocytes from TGF-βR-KO mice and TGF-βR-WT mice; the histogram demonstrates the percentage of HSA[low] BIM[neg] among the Foxp3[neg] Helios[pos] CD4 SP thymocytes. **e** Flow cytometry analysis for surface expression of S1P1 on Foxp3[neg] Helios[pos] CD4 SP thymocytes, histogram illustrates the S1P1 MFI in Foxp3[neg] Helios[pos] CD4 SP thymocytes. **f** *CD4-Cre;Tgfbr2[fl/fl]* mice were injected intra-thymically with FITC and their peripheral lymph nodes were analyzed 24 h later by flow cytometry. Histogram plot illustrates the proportion of FITC[pos] CD4[pos] T lymphocytes (top) and was used as gating strategy for the expression of Helios among Foxp3[neg], CD4[pos] FITC[pos] T lymphocytes. The graph demonstrates percentage of Helios[pos] among CD4[pos] FITC[pos] T lymphocytes. All the experiments were conducted in 7–10-day-old animals. Data are representative of 2–3 experiments, with 3–5 mice per group. *P < 0.05, **P < 0.01, ***P < 0.001 (two-tailed Student's t test). Error bar, mean ± SEM. Source data are provided as a Source Data file.

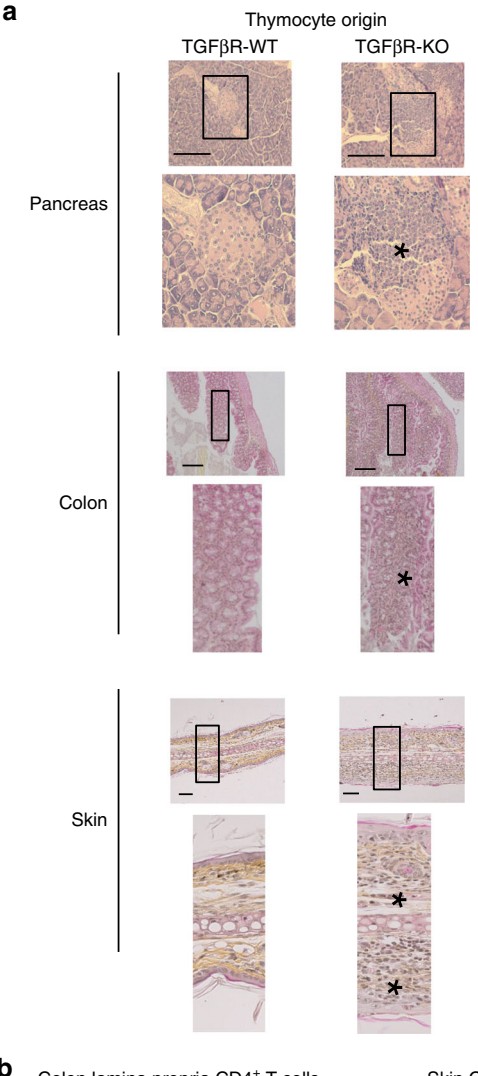

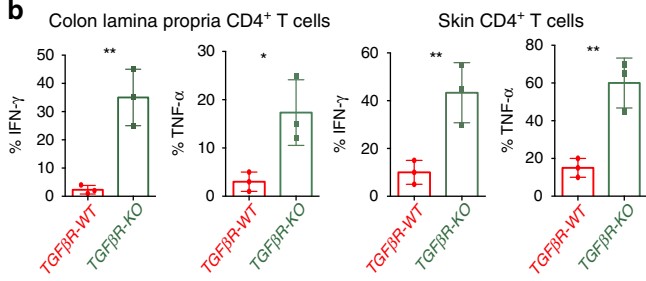

**Fig. 6 Mature thymocytes from TGFβR-KO mice lead to multiple organ autoimmunity.** CD3-KO mice were grafted with FACS-sorted CD4 SP (CD69⁻, CD24^Low, Qa2⁺) thymocytes from either TGFβR-WT mice or TGFβR-KO mice. Recipients and donors were 7–10 days old on the day of the transfer; 8 weeks after being grafted, the recipients were analyzed. **a** Representative micrographs of H&E-stained tissue sections of indicated organs; * indicates the infiltrates. Data are representative of 2 experiments with 3 animals per group. Scale bars, 50 μm (magnification 3×). **b** Cells infiltrating the lamina propria of the colon as well as the skin were isolated and analyzed by flow cytometry. Graphs demonstrate the percentage of CD4⁺ T cells producing IFN-γ or TNF-α. Data are representative of 2 experiments with 3 animals per group. **P < 0.01 (two-tailed Student's t test). Error bar, mean ± SEM. Source data are provided as a Source Data file.

**Table 1 Frequency of the observed pathologies.**

| | Thymocyte origin | |
| --- | --- | --- |
| | **TGF-βRWT** | **TGF-βRKO** |
| Mice with infiltrated organs | 15% (1/7) | 100% (7/7) |
| Mice with more than one organ affected | 0% (0/7) | 100% (7/7) |
| Mice with colon infiltration | 15% (1/7) | 100% (7/7) |
| Mice with insulitis | 0% (0/7) | 80% (6/7) |
| Mice with dermatitis | 0% (0/7) | 80% (6/7) |
| Mice with liver infiltration | 0% (0/7) | 60% (5/7) |
| Mice with brain infiltration | 0% (0/7) | 20% (2/7) |

CD3-KO mice were grafted with FACS-sorted CD4 SP (CD69⁻, CD24^Low, Qa2⁺) thymocytes from either TGFβR-WT mice or TGFβR-KO mice. Recipients and donor were 7–10 days old on the day of the transfer; 8 weeks after being grafted, the recipients were analyzed. The frequency of the observed pathologies is illustrated in % and number of mice are indicated

cortex and in the medulla remains to be determined. It was proposed that apoptosis generated by negative selection triggers TGF-β1 production in the thymus, which in turn drives Foxp3 expression in thymic Treg (tTreg) precursors[34]. Interestingly, the initiation of tTreg production, whose first wave of development occurs normally at around day-2-3 after birth in TGF-β signaling sufficient animals[35], was postponed to day-7-8 in TGFβR-KO mice[36]. Thus, our work suggests that the defect in negative selection in TGFβR-KO mice could contribute to the delayed development of their tTreg. However, this latter explanation cannot explain the massive multiple organ autoimmunity that occurs in these animals, since normal development of wild-type tTreg in WT:KO BM chimera mice failed to cope with the development of autoimmune lesions[18,19,36]. Furthermore, peripheral autoimmune manifestations are more drastic in TGFβR-KO mice compared to those seen in Aire-KO mice on a same genetic background[37]. This is likely due to the fact that not only medullary but also cortical negative selection was severely impaired in TGFβR-KO mice. Indeed, cortical negative selection, which is independent of AIRE (Fig. 2g), has been estimated to eliminate more autoreactive thymocytes (~75% at the DP stage) than medullary negative selection (~25% at the SP stage)[2,3,38]. In addition, TGF-β signaling can modulate TCR activation thresholds[39] and thus increase activation of the high number of autoreactive T lymphocytes that massively exit the thymus in TGFβR-KO mice.

Once bound to its receptor, TGF-β can signal through different pathways[16]. Our data incriminate the non-canonical branch of TGF-β signaling in the negative selection, assigning a multifaceted role in the process of negative selection. We revealed that, during cortical negative selection, TGF-β signaling, via its non-canonical pathway, directly promotes BIM expression in highly autoreactive thymocytes. A pro-apoptotic role of the non-canonical branch of TGF-β signaling has been shown in numerous cell types, and can involve a transcriptional regulation by rapid inhibition of ERK that prevents the phosphorylation and degradation of BIM[40]. In line with this, *Bim* mRNA levels were similar between Foxp3^neg Helios^pos CD4^pos CD8^low thymocytes from TGF-βR-KO mice and TGFβR-WT mice (Supplementary Fig. 6) suggesting that the non-canonical branch of TGF-β signaling sustains the high levels of BIM, induced in response to strong TCR affinity to self-antigen MHC complexes in the cortex.

Interestingly, TGF-β signaling also prevents highly autoreactive thymocytes from acquiring a "Th1 like" phenotype with the

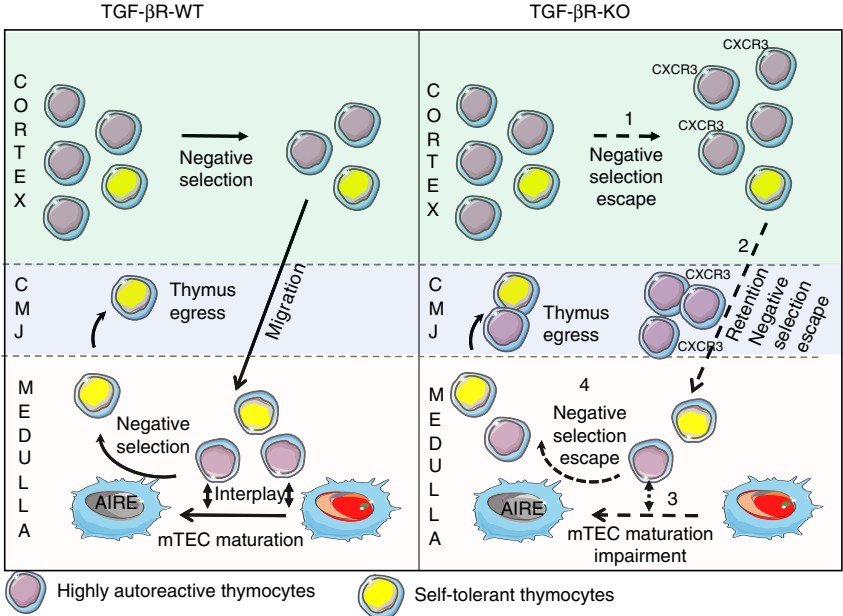

**Fig. 7 Schematic overview of the multiple effects of TGF-β on negative selection.** In the absence of TGF-β signaling, highly autoreactive thymocytes escape cortical negative selection (**1**). They then acquire CXCR3 at their surface and a large fraction of them are trapped at the CMJ (cortico-medullary junction) (**2**), preventing their entrance in the medulla where a second wave of negative selection occurs. The few highly autoreactive thymocytes entering the medulla fail to condition mTECs for an efficient negative selection: AIRE[pos] mTEC differentiation is largely altered as well as their ability to express TRA (**3**). Highly autoreactive thymocytes escape the second wave of negative selection (**4**). In sum, in the absence of TGF-β signaling control in thymocytes, several mechanisms essential for elimination of highly autoreactive thymocytes are impaired in the thymus allowing them to reach the periphery where they can induce tissue damage. The figure has been created by J.C.M.

expression of T-Bet leading to CXCR3 expression, a chemokine receptor usually not expressed by thymocytes, except by iNKT thymocytes that also remain at the CMJ[25]. As for iNKT cells, the expression of T-Bet in Foxp3[neg] Helios[neg] CD4 thymocytes is also repressed by the non-canonical pathway of TGF-β signaling[41]. Whether the CMJ constitutes a specific niche with peculiar signals provided to thymocytes settled there remains to be determined. However, in TGFβR-KO mice, the weak proportion of BIM[pos] cells among the CXCR3[pos] Foxp3[neg] Helios[neg] CD4 thymocytes implies that cells that present at the CMJ fail to purge highly autoreactive cells. This idea is in agreement with the large defect in TGFβR-KO mice of AIRE-dependent TRAs that are acquired after mTEC apoptosis by other thymic cells and particularly DCs after engulfment of apoptotic bodies[42]. Notably, the impairment of AIRE[pos] mTEC differentiation in the absence of TGF-β signaling in thymocytes is at least the conjunction of two effects namely the reduction in levels of RANKL on highly autoreactive Foxp3[neg] Helios [pos] thymocytes and the impaired ability of these cells to properly reach the medulla and thus deliver to them signals (Fig. 7), including RANK and CD40 signaling, essential for their maturation[24]. Though, we cannot address the specific contribution of these two effects and exclude additional factors that could prevent AIRE expression in mTEC, our observations (Fig. 4b) showed that the RANKL expressio n by Helios[neg] thymocytes was not sufficient to sustain AIRE[pos] mTEC differentiation in TGFβR-KO mice.

Several studies have shown that TGFβR-KO mice develop severe autoimmunity that is characterized by a robust Th1 response[18,19]. Importantly, one study reported that in TGFβR-KO mice, negative selection was not impaired but exaggerated implying that autoimmunity observed in these animals would be due to a loss of peripheral tolerance mechanisms rather than a defect in thymic negative selection[43]. However, the ablation of TGF-β signaling in peripheral mature T lymphocytes fails to

induce autoimmunity[21,22]. The discrepancy between Ouyang et al.[43] and our study could be explained by the choice of the methods to investigate negative selection. Indeed, anti-CD3 antibody injections, which lead to BIM-independent apoptosis[44], were used in Ouyang et al., and a 12-hour-culture was-required to observe the first annexin V positive staining.

In sum, with a clear reduction in BIM expression in highly autoreactive thymocytes both in the cortex and in the medulla, as well as defective AIRE[pos] mTEC differentiation and TRA expression, and the development of massive autoimmune manifestations after adoptive transfer of thymocytes from TGFβR-KO mice, we believe that our work firmly establishes an unrecognized function for TGF-β in thymocyte negative selection. This study improves our knowledge on the role played by TGF-β within the immune system and may constitute a turning point for future immunotherapies of autoimmune diseases and cancer.

## Methods

**Mice.** *CD4-Cre;Tgfbr2[fl/fl]* mice[19] and *CD4-Cre;Smad4[fl/fl]* mice, *CD4-Cre;tif1g[fl/fl]* mice and *CD4-Cre; tif1g[fl/fl]; Smad4[fl/fl]* mice (C57BL/6 background) described in ref. [41] were bred with *Foxp3[GFP]* mice kindly given by A.Y. Rudensky (Sloan Kettering Institut, USA) to eliminate Foxp3[pos] cells in gating strategies. AIRE-KO mice were kindly provided by B. Kyewski (DKFZ, Germany). Rag2-deficient mice (Rag-KO), CD3-KO mice, Thy1.1 (CD90.1), CD45.1 mice were purchased from Charles River Laboratories. In experiments using Cre expressing strains, Cre-only expressing mice were used as wild-type littermate controls, since Cre can influence the thymus in the absence of floxed genes[23]. No noticeable difference was observed between genders in these strains. Mice were housed and bred in a specific pathogen-free animal facility, AniCan, at the C R C L.

**Ethics.** Experiments on mice were performed in accordance with the animal care guidelines of the European Union and French laws and were validated by the local Animal Ethic Evaluation Committee (CECCAPP).

**Cell isolation**. Thymocyte suspensions were prepared by manual disruption using glass slides. When mentioned, Foxp3[pos] cells were excluded based on GFP expression. BM were flushed from leg bone and depleted of CD3[pos] T cells using anti-CD3 magnetic beads (Miltenyi-Biotec). For mTEC isolation, thymic fragments were digested in PBS containing 1 mg/ml collagenase D (Sigma-Aldrich) and 2 µg/ml DNaseI (Roche). Remaining hematopoietic cells of the preparation were removed with anti-CD45 magnetic beads (Miltenyi Biotec). Both mTEC and thymocyte purification was performed by cell sorting on a FACSAria II (BD biosciences). For lamina propria, and skin, organs were was cut into small pieces and incubated with 5 mM EDTA, 1 mM DTT (Sigma-Aldrich) at 37 °C. Epithelial cells were separated from tissue. Tissues were then digested with 0.6 mg/ml of collagenase D (Sigma-Aldrich) and 100 µl of DNase I (Roche) and cells centrifuged on a 40% Percoll gradient (Fisher Scientific).

**Fetal thymic organ culture**. Fetal thymus lobes were dissected from E17.5 embryos and cultured for 4 days on sponge-supported filter membranes (Gel Foam surgical sponges; Amersham Pharmacia, Piscataway, NJ; Supor 450 membrane, 0.45-µm pore size, Millipore Merck) at an interphase between 5% $CO_2$-humidified air and completed RPMI medium 1640, 10% FCS, 10 mM HEPES (Life Technology), and penicillin streptomycin (Thermo Fisher).

**Antibodies and flow cytometry**. Cells were pre-incubated with anti-CD16/32. Surface staining was performed using the following fluorescent-conjugated antibodies:, CD3 (145-2C11; BD Biosciences), CD4 (RM4-5; Biolegend), CD5 (53-7.3; BD Biosciences), CD8α (53.6.7; BD Biosciences), CD24 (M1/69; BD Biosciences), CD40L (MR1; eBioscience), CD45.1 (A20; BD Biosciences), CD45.2 (104; BD Biosciences), CD45 (30-F11; BD Biosciences), CD69 (H1.2F3; BD Biosciences), CD80 (16-10A1 eBioscience) CD90.2 (53-2.1; BD Biosciences), CD90.1 (Ox7 BD Biosciences), CCR7 (4B12; BD Biosciences), CXCR3 (CXCR3-173; eBioscience), EpCAM (G8.8; eBioscience), H2K^b (AF6-88-5.5.3 eBioscience) LY51 (6C3/BP-1; BD Biosciences), PD1 (EH12.2H7; BioLegend), S1P1 (T4-H28; R&D systems), Qa2 (1-1-2 BD Biosciences) TCRαβ (H57–597; BD Biosciences). For cytokine staining, cells were incubated in brefeldin A (eBioscience), for 3 h. RANKL (IK22-5; BD Biosciences), IFN-γ (XMG1.2 BD Biosciences), TNF-a (MP6XT22 BD Biosciences) were performed with cytofix/cytoperm kit (BD Pharmingen). For activated caspase 3 (C92-605; BD Biosciences), and BIM (C34C5; Cell Signaling Technology), cells were fixed and permeabilized using the BD cytofix/cytoperm kit (BD Pharmingen). For AIRE (5H12; eBioscience), Foxp3 (FJK-16S; eBioscience), Helios (22F6; BioLegend), Ki67 (B56; BD Biosciences), and T-Bet (eBio4B10; eBioscience) stainings, cells were fixed and permeabilized using the intra-nuclear staining kit according to manufacturer's instructions (eBioscience). Flow cytometry analyses were carried out using a BD Fortessa (BD Biosciences) and analysis of cells was performed using FlowJo software (Miltenyi). Fluorescence minus one (FMO) controls were used to set positive staining gates.

**Immunofluoresence staining and histological analysis**. Thymus were embedded in Tissue-Tek OCT compound (Sakura Finetek) and snap-frozen over liquid nitrogen; 20-µm-thick sections were cut with a Leica X cryostat. Anti-rat MTS10 (Pharmingen) revealed with Alexa Fluor 488-conjugated anti-rat, rabbit anti-K14 (AF64, Covance Research) revealed with Cy3-conjugated anti-rabbit, Alexa Fluor 488 anti-AIRE (5H12, ebioscience), Alexa Fluor 647-conjugated anti-Helios (22F6, eBioscience) and Alexa Fluor 488-conjugated anti-Foxp3 (eBioscience) were used. Sections were permeabilized for 10 min in 0.1 M Tris buffer pH 7.4, 0.02% Triton X-100 (Sigma-Aldrich). Before incubation with primary antibodies, sections were incubated for 10 min in saturation buffer composed of 0.1 M Tris buffer pH 7.4, 3% BSA (Axday), 0.01% Triton X-100. Sections were counterstained with DAPI (1 µg/ml) and mounted with Mowiol (Calbiochem). Quantification was performed with ImageJ software. Tissue sections (5 µm) were prepared with 4% paraformaldehyde fixed, paraffin embedded tissues and stained with hematoxylin–eosin (H&E) with standard protocols. Immunofluorescence staining was observed on a LSM 780 Leica SP5X confocal microscope and tissue sections were observed on a Zeiss Imager M2 microscope. Helios[pos] Foxp3[neg] cellularity in the CMJ was automatically computed with Matlab (The Mathworks) as follows: the medulla was detected by K14 staining, in the red channel. The image was first smoothed by a 15-pixel median filter (0.31-µm/pixel calibration). The resulting image was thresholded using the maximum entropy method (function kindly provided by F. Gargouri, Mathworks File Exchange), then smoothed by dilatation over 95 pixels and erosion over 80 pixels. Objects with an area smaller than 700 pixels were discarded. The resulting binary image provided the inner border of the CMJ. The outer border was computed by further dilating the CMJ over 50 µm (160 pixels). Helios[pos] and Foxp3[pos] cells were detected from the cyan and yellow channels, respectively, using a similar scheme: images were soothed by a median filter of 3-pixel width. The resulting images were thresholded using the moment method (function kindly provided by D. Martín (ref. [45])). Objects with an area smaller than 15 pixels were discarded. Thresholds were validated by visual inspection. Helios[pos] Foxp3[pos] cell density was computed as the number of cells within the CMJ, divided by the CMJ area. Medullary areas were computed with Matlab as follow: for each image, the thymus was detected from the DAPI channel. The image was smoothed by a 3-pixel median filter (26-µm/pixel calibration). The

| **Table 2.** | |
|---|---|
| **Primers** | **Sequence 5′->3′** |
| *Gadph* forward | GCATGGCCTTCCGTGTTC |
| *Gadph* reverse | TGTCATCATACTTGGCAGGTTTCT |
| *Actin* forward | CAGAAGGAGATTACTGCTCTGGCT |
| *Actin* reverse | GGAGCCACCGATCCACACA |
| *Rank-L* forward | CGCTCTGTTCCTGTACTTTC |
| *Rank-L* reverse | AGAGTCGAGTCCTGCAAATC |
| *Insulin* forward | AGCCCTAAGTGATCCGCTAC |
| *Insulin* reverse | GCATCCACAGGGCCATGTTG |
| *Sp1* forward | CTGGTGAAAATACTGGCTCTGAA |
| *Sp1* reverse | GCAGTGTTGGTATCATCAGTG |
| *Sp2* forward | TCAGACCAAAGTGGGTGACA |
| *Sp2* reverse | CCTCTTGTTTCTCATTGGAGGT |
| *Mup1* forward | ATTCTGTGACGTATGATGGA |
| *Mup1* reverse | GAAGTCTCCACTCAACACTG |
| *β casein* forward | CTCCACTAAAGGACTTGACAG |
| *β casein* reverse | ACCTTCTGAAGTTTCTGCTC |
| *Bim* forward | CGGATCGGAGACGAGTTCA |
| *Bim* reverse | TTCCAGCCTCGCGGTAATCA |

resulting image was thresholded using the Otsu method (ref. [46]), then smoothed by 3-pixel dilatation and erosion. The thymic contour was eroded by 12 pixels. The staining was analyzed only in this eroded area, to remove putative non-specific staining at the thymic capsule[47]. The medulla was filtered over 3 pixels, thresholded using the moment method and smoothed over 3 pixels. Surfaces were computed for the resulting images. Since we established that medullary topology is isotropic[47,48], medullary surfaces analyzed for medial sections directly represent medullary volumes for whole thymi.

**In vivo treatments**. For BM transfer, $1 \times 10^6$ donor BM-derived cells were injected intravenously into 6-8-week-old Rag2-KO recipient mice that were previously irradiated (6 Gy). For thymocyte injection, $5 \times 10^6$ sorted CD69[neg] HSA[low] CD4[pos] CD8[neg] thymocytes were injected intravenously into 2–3-week-old CD3-KO recipient mice. Intra-thymic injections were performed using ultrasound monitoring (VivoLazer, VisualTonic) on 10-day-old mice with 5–8 µl of a solution of 3 mg/ml FITC (Sigma) in PBS.

**Culture and real time PCR analysis**. For the monitoring of *Rank-l* expression, $5 \times 10^5$ FACS-sorted Foxp3-GFP[neg] CD4[pos] CD8[low] PD1[pos] thymocytes from Foxp3-GFP; TGFβR-WT mice were settled for 1 h in PBS and cultured for 3 h in RMPI medium supplemented with 200 mM L-Glutamine, 10 mM HEPES at 37 °C and 5% of $CO_2$ in 96-well plates previously coated overnight with anti-CD3 (2C11, BD Pharmingen) at 1 µg/ml in PBS. 5 ng/ml human recombinant TGF-β1, 1 mM TGF-βR1 inhibitor (SB431542) and 1 mM p38 MAPK inhibitor (SB203580) were used (R&D system).

mRNA from cultured thymocytes and FACS-sorted mTEC (EpCAM[pos] Ly51[low] CD45[neg]) were isolated with RNeasy mini kit (Qiagen) and reverse transcribed with iScript cDNA synthesis kit (Bio-rad). Real-time RT-PCR was performed using LightCycler 480 SYBR Green Master (Roche) and different sets of primers on LightCycler 480 Real-Time PCR System (Roche). Samples were normalized on either *gadph* expression (Thymocytes) or *actin* expression (mTEC) and analyzed according to the ΔΔCt method (Table 2).

**Statistics**. Unpaired two-tailed Student's *t* test was used to assess the significance of differences observed between the two groups with Prism software. *P* values < 0.05 were considered significant.
    *$P < 0.05$, **$P < 0.001$, ***$P < 0.0001$.

**Reporting summary**. Further information on research design is available in the Nature Research Reporting Summary linked to this article.

## Data availability
Data supporting the findings of this study are provided either by supplementary figures or in The Source data file (data for Fig. 1d; 2a, c–f; 3b, d; 4b, d; 5a–c; 6b, and Supplementary Figs. 1; 2; 3a, b; 5; 6 are provided in the Source data file).

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

## Acknowledgements

The expert assistance of V. Ballaz, J. Noiret and C. Deceneux is acknowledged. We also thank all the Marie lab members for their helpful discussions as well as the staff of the core facilities at the Cancer Research Center Lyon (AniCan, CytoCan) for their technical assistance. Profs. Rudensky, Kyewski, Dardalhon and Grinberg-Bleyer are warmly acknowledged for providing mice and advice. This work was supported by grants from the Helmholtz-DKFZ-Inserm program (JCM), Agence Nationale pour la Recherche (ANR) investissement d'avenir ANR-10-LABX-61 (JCM), the foundation Schuller-Bettencourt (JCM), Ligue Nationale Contre le Cancer labelisée, EL-2016, Marie Curie Career integration grant CIG_ SIGnEPI4Tol_618541 (MI) and the Swiss National Science Foundation PZ00P3-131945 (MI). M.J.M.C. was supported by the Fondation pour la Recherche Medicale; SPF20110421356. J.C.M. was a Helmholtz-association investigator.

## Author contributions

M.J.M.C., M.I., S.M.S. and J.C.M. planned and conducted the experiments, analyzed the data and wrote the manuscript. A.S. performed images analysis and quantification. J.C.M. supervised the study.

## Competing interests

The authors declare no competing interests.
