## [Peer Review File · Nature Communications]

Reviewers' comments:

Reviewer #1 (Remarks to the Author):

In this paper by McCarron et al, the authors outline a key role for TGFb in ensuring appropriate thymic negative selection. The authors present data showing impaired negative selection in the absence of TGFb signaling and show that surviving thymocytes accumulate at the CMJ and fail to appropriately stimulate mTECs to sustain AIRE expression. The data presented by the authors describes an important new function for the cytokine TGFb within the thymic environment and would be of interest to a substantial number of scientists. However, a few questions/issues should be addressed before publication. (none of the issue raised related to statistical analysis all of which is appropriate within the manuscript).

Major Comments:

- 1) The authors present data showing that TGFb-KO thymocytes cannot promote/support AIRE expression in mTECs (figure 2b-e). However, it remains possible that the defects in the mTEC seen in the TGFbR-KO mice are due to some inflammatory side effect (as occurs in the TGFbR1-CD4cre mice) or other mechanism. As such it would be important to show what happens to the mTECs in chimera mice, for example those shown in Figure 2a. Does the presence of WT thymocytes rescue the defects in mTEC that are seen when only TGFbR-KO thymocytes are present?
- 2) The authors suggest that in the absence of TGFb signals thymocytes position incorrectly within the thymus due to CXCR3 expression. To show that this is an intrinsic defect (and for example not due to the aberrant inflammation – a key point when considering the mouse they are working with), the authors should intra-thymically inject the different populations of CD4 SP thymocytes from WT and KO mice and see where they localize in a non-inflamed thymus.
- 3) Given the data presented and the central points about escape from negative selection the transfer experiment shown in figure 6 is important but needs refinement to demonstrate the points stated by the authors. Instead of transferring bulk CD4 SP cells the authors should transfer the Helios+foxp3-subpopulation that have escaped negative selection. (This reviewer appreciates that helios in an intracellular stain but perhaps PD-1 or a combination of CD69, MHCII, Qa2 and/or CXCR3 could be used). This is important to show, especially as it would also be possible to demonstrated whether these cells mediate autoinflammation faster, to a greater extent etc etc that other populations of thymocytes also sorted from the TGFbR-KO transgenic mouse.

Minor Comments:

- 1) On page 9 on the manuscript the authors mention that no differences were seen in CD154 expression on thymocytes between WT and KO mice. This data should be shown.
- 2) On page 10 the authors mention that in mice lacking SMAD4 and TRIM33 or both had normal RANKL expression on thymocytes. Again this data should be shown.

Reviewer #2 (Remarks to the Author):

This manuscript describes that CD4-Cre-mediated deletion of TGF-beta-receptor affects Helios-high thymocytes, Aire-expressing mTECs, and downstream T cell development. However, how the findings reported in this manuscript are incorporated into our understanding regarding the role of TGF-beta signals in the thymus, which have already been shown in various aspects including Treg generation and DP thymocyte development. The thymic negative selection, which is defined exclusively by Helios-high expression in this manuscript, should be better defined in more conventional methods.

Reviewer #3 (Remarks to the Author):

The manuscript from McCarron et al., "TGF-beta signaling in alpha/beta thymocytes promotes negative selection" proposes that this growth factor is required for the deletion of autoreactive T cells during their differentiation in the thymus. The authors show that mice lacking TGF-beta receptors accumulate aberrant populations of thymocytes with features consistent with autoreactive cells and that the microenvironment lacks key AIRE+ medullary epithelial cells. These new data suggest a novel mechanism for thymic deletion induced by TGF-beta that would be of interest to immunologists in the field of T cell differentiation and immune tolerance. However, I have several major concerns about these findings that preclude me from recommending publication in Nature Communications in the current form.

MAJOR POINTS:

Could the aberrant population of CD4+CD8low HELIOShigh FOXP3neg cells represent recirculating activated T cells? I note that the authors assay the mice at early timepoints; however, the inflammatory phenotypes develop early in these mice and it is conceivable that substantial peripheral activation has already occurred. The phenotypic features presented are consistent with this interpretation. Alternatively, could they be iNKT cells or IELp? A more comprehensive phenotypic analysis and comparison with peripheral T cells would help provide support for the author's conclusions. Previous studies have used a pulse of EdU to track a cohort of nascent thymocytes through their differentiation (e.g. Daley, Hu and Goodnow, JEM) which could be applied here to establish the kinetics of this population. In addition, if these aberrant cells arose in fetal thymic organ culture, it would provide very strong evidence that this phenotype does not reflect recirculating activated T cells. Testing this possibility is an important requirement for supporting the author's primary conclusion.

The link between impaired induction of AIRE+ mTEC, the aberrant T cell population and autoimmunity is a potentially important finding, but several aspects of this element of the study are puzzling. Firstly, although RANKL levels are clearly reduced in the aberrant HELIOSpos FOXP3neg cells, the overall proportion of thymocytes expressing this ligand is comparable between WT and KO mice based on the dot plots in Figure 4A (proportions in the upper quadrants). Therefore, TGFb cannot be necessary for RANKL expression, nor can this mechanism be the complete explanation for why AIRE+ cells are not being induced. Are the CD4posCD8low cells that are HELIOShigh FOXP3neg producing something that interferes with RANK signaling in mTEC? Also, none of the autoimmune features shown in Figure 6 are normally observed in AIRE KO mice (I'm presuming here that all the mice used in this study are on an inbred C57BL/6 background – this is not stated in materials). Further development of this line of enquiry is essential to support this potentially interesting finding.

MINOR POINTS:

SuppFig 2; There is no measure of medullary volume. Please add quantification of this region to support the conclusion that the size of the medulla is unaffected in the TGFbRKO. It actually looks like there might be a more subtle impact, with many more small islets of medulla apparent in the KO compared to the WT. Is the localization of SP thymocytes normal?

There is a very large fraction of active caspase-3+ TEC, which is highly unlikely to represent the true rate of apoptosis in this population in vivo. Most of this cell death is likely to have been induced by the sample preparation. I suggest leaving these data out because they are not an adequate test of the hypothesis and not essential to the key finding that AIRE+ cells are reduced.

Figure 3. The cortico-medullary junction (CMJ) is being defined as a 100um region extending from the K14+ mTEC boundary. Firstly, the zones indicated in Figure 3A seem rather variable (i.e. do not look

like a uniform 100um from the K14 boundary). More detail of how the analysis of the CMJ were performed using ImageJ is required. Secondly, given that the authors have already established a defect in the maturation of AIRE+ mTEC (many of which reside near the CMJ), is it suitable to base this distinction using a mTEC marker like K14? A more suitable definition of the CMJ would be based on a cortical epithelial marker (e.g. beta 5t or K8), which may be normal in the TGFbRKO mice.

The conclusion that the aberrant CD4posCD8low HELIOShigh cells are prevented from export requires further support. Maturing thymocytes are able to leave the thymus without going to the medulla (e.g. CCR7KO mice). Could it be that these are activated T cells that have come into the thymus, and localized at the CMJ due to their Th1-like phenotype and expression of CXCR3?

Do the CD4posCD8low cells that are HELIOShigh FOXP3neg and the CD4SP HELIOShigh cells express CCR7? Hard to tell from the histograms of the bulk population shown in 1B, and might be better shown by gating directly on these populations to assess CCR7 expression. This will provide important data on potential location. Likewise, what is the expression of CXCR4 (which is critical for the cortical localization of DP) and S1P1 (required emigration; this is shown for CD4SP in Fig 5e but not his population).

Figure 5f. The recent thymic emigrant experiment shown here is a good one, but the authors restrict their analysis to only HELIOS expression among the RTE. They should show a broader comparison of the overall FITC+ RTE in WT vs KO. Are there more RTE in the KO? What is their predominant phenotype? These data might provide more insight into thymic function and the preferential export of cells between the genotypes.

Figure 6. Some quantification of the autoimmune phenotypes observed is essential. Did all mice show these infiltrates? How severe were they? For instance, the images of pancreas shown demonstrate some insulinitis, but there does not appear to be destruction of the islet.

Point by point answers to the reviewers

Reviewer #1

We thank the Reviewer #1 for his/her time and the useful comments to improve the original version of our work. We also appreciate that Reviewer #1 underlined that our data describes an important new function for the cytokine TGF- β within the thymic microenvironment and that our work would be of interest to a substantial number of scientists.

1) *The authors present data showing that TGF β -KO thymocytes cannot promote/support AIRE expression in mTECs (figure 2b-e). However, it remains possible that the defects in the mTEC seen in the TGF β R-KO mice are due to some inflammatory side effect (as occurs in the TGF β R1-CD4cre mice) or other mechanism. As such it would be important to show what happens to the mTECs in chimera mice, for example those shown in Figure 2a. Does the presence of WT thymocytes rescue the defects in mTEC that are seen when only TGF β R-KO thymocytes are present?*

We totally understand the possibility raised by the Reviewer #1, questioning a potential side effect of the inflammation observed in TGF β R-KO mice on mTEC differentiation. Bone marrow (BM) chimera mice (TGF β R-WT: TGF β R-KO) were previously reported to develop inflammation similarly to that observed in TGF β R-KO mice (Li et al Immunity 2006, Marie et al Immunity 2006). We analyzed AIRE expression in mTEC in the BM chimera experiment suggested by the reviewer #1. We found that the presence of TGF β R-WT thymocytes rescued the defective maturation of mTEC observed when only TGF β R-KO thymocytes are present. We thus believe that these data, shown in figure 2f, rules out any side effects of inflammation on the defect in AIRE^{Pos} mTEC differentiation and thanks the reviewer for the suggestion. The text on page 8 has been modified to underline this point.

2) *The authors suggest that in the absence of TGF β signals thymocytes position incorrectly within the thymus due to CXCR3 expression. To show that this is an intrinsic defect (and for example not due to the aberrant inflammation – a key point when considering the mouse they are working with), the authors should intra-thymically inject the different populations of CD4 SP thymocytes from WT and KO mice and see where they localize in a non-inflamed thymus*

We agree with Reviewer #1 that to know whether the expression of CXCR3 on thymocytes is regulated intrinsically by TGF- β signaling or by extrinsic factors is of importance. We appreciate the suggestion made by the Reviewer to perform intra-thymic injection of thymocyte populations in order to address this question. However, to our knowledge, there is no imaging tool to inject thymocytes in a precise compartment of the thymus, i.e. cortex, medulla or CMJ. Hence, we believe that this challenging approach will not allow drawing firm conclusions without knowing if SP thymocytes were injected into the medulla or at the CMJ, or even in the cortex.

With respect to addressing the concern of the reviewer, we performed another approach to determine the intrinsic role of TGF- β signaling on the expression of CXCR3. We analyzed the expression of CXCR3 in thymocytes from BM chimera mice (TGF β R-WT: TGF β R-KO) known to develop inflammation similarly to TGF β R-KO mice (Li et al Immunity 2006, Marie et al Immunity 2006). In clear contrast to TGF β R-KO thymocytes (CD45.2^{Pos}), TGF β R-WT thymocytes (CD45.2^{neg}) expressed extremely weak levels of CXCR3 (15% versus 45% for Foxp3^{neg} Helios^{pos} CD4^{pos} CD8^{low} and 9% versus 40% for Foxp3^{neg} Helios^{pos} CD4 SP), confirming the intrinsic effect of TGF- β signaling on CXCR3 expression on Helios^{pos} thymocytes. These new data are now shown in supplementary figure 4 and the importance of

the intrinsic control of CXCR3 by TGF- β has been underlined in the text of the revised manuscript (pages 9).

3) Given the data presented and the central points about escape from negative selection the transfer experiment shown in figure 6 is important but needs refinement to demonstrate the points stated by the authors. Instead of transferring bulk CD4 SP cells the authors should transfer the Helios+foxp3- subpopulation that have escaped negative selection. (This reviewer appreciates that helios in an intracellular stain but perhaps PD-1 or a combination of CD69, MHCII, Qa2 and/or CXCR3 could be used). This is important to show, especially as it would also be possible to demonstrated whether these cells mediate autoinflammation faster, to a greater extent etc etc that other populations of thymocytes also sorted from the TGF β R-KO transgenic mouse.

We thank the Reviewer #1 for underlining the importance of the message delivered by the data shown in figure 6. We really appreciate the willingness of Reviewer #1 to suggest experiments to show that Helios^{pos} Foxp3^{neg} thymocytes are incriminated in the autoimmunity reported in Figure 6. However, we agree with Reviewer #1 that to sort Helios^{pos} thymocytes is impossible, as Helios is an intracellular protein. Moreover, as sensed by the Reviewer #1, there are unfortunately no good indirect sorting strategies. Indeed, sorting Foxp3-GFP^{neg} PD-1^{pos} thymocytes will lead us to transfer only immature DP^{low} thymocytes (Daley et al J. Exp Med 2012). The use of CXCR3 as a marker of Helios^{pos} cells is not accurate since CXCR3 is also expressed by a fraction of Helios^{neg}, as shown in Figure 3c. In the same vein, sorting Foxp3^{neg} CD69^{neg}, MHC-I^{high}, QA2^{pos} thymocytes will lead us to inject all mature thymocytes regardless of their Helios expression as shown in Figure5a which is what was actually done in figure 6. Thus, as appreciated by the reviewer, we do not believe that there is an experimental strategy to address the point raised besides the generation of TGF β R-KO mice with a Helios RFP reporter allele, which will take around 2 years to get all the transgene expressed (CD4-Cre, TGF β R2^{fl/fl}, Foxp3-GFP) in the newly generated mice.

We totally agree with the Reviewer #1 that a formal demonstration that Helios^{pos} thymocytes mediate faster tissue auto-inflammation will be of interest. However, it is technically impossible. As set out in the title of our article, the scope of our manuscript is: “the defect of thymic negative selection in TGF β R-KO mice”, which is illustrated by (i) survival of highly autoreactive thymocytes in the un-skewed repertoire, (ii) the defect in cortical negative selection, (iii) the defect of AIRE+ mTEC and TRA expression, (iv) the defect in medullary negative selection. The autoimmune symptoms observed after transfer of TGF β R-KO thymocytes at the periphery help enforced our message, by providing a pathological consequence of negative selection impairment in TGF β R-KO mice

Minor comments:

4) On page 9 on the manuscript the authors mention that no differences were seen in CD154 expression on thymocytes between WT and KO mice. This data should be shown.

We apologize for not including the data as a figure. Supplementary Figure 5 has been created to show CD40L (CD154) surface expression. As the Reviewer knows CD40L being largely internalized after contact with APC, its surface expression is always weak on thymocytes.

5) On page 10 the authors mention that in mice lacking SMAD4 and TRIM33 or both had normal RANKL expression on thymocytes. Again this data should be shown.

We apologize for mentioning as data not shown RANKL expression in the aforementioned mice. These data showing the absence of an effect of the deprivation of these two actors of TGF- β signaling has been added in figure 4a as requested.

Reviewer #3

We thank the Reviewer #3 for reinforcing the novelty of our work and its interest to immunologists in the fields of T cell differentiation and immune tolerance. We also understood that Reviewer #3 had several concerns that we addressed by providing additional data based on her/his suggestions.

1) Could the aberrant population of CD4+CD8low HELIOShigh FOXP3neg cells represent recirculating activated T cells? I note that the authors assay the mice at early timepoints; however, the inflammatory phenotypes develop early in these mice and it is conceivable that substantial peripheral activation has already occurred. The phenotypic features presented are consistent with this interpretation. A more comprehensive phenotypic analysis and comparison with peripheral T cells would help provide support for the author's conclusions. Previous studies have used a pulse of EdU to track a cohort of nascent thymocytes through their differentiation (e.g. Daley, Hu and Goodnow, JEM) which could be applied here to establish the kinetics of this population. In addition, if these aberrant cells arose in fetal thymic organ culture, it would provide very strong evidence that this phenotype does not reflect recirculating activated T cells. Testing this possibility is an important requirement for supporting the author's primary conclusion.

Though BM chimera analysis illustrated in figure 2a suggested that the disruption of Foxp3^{neg} Helios^{pos} TGFβR-KO thymocyte homeostasis was not due to a side effect of the inflammation that occurs in TGFβR-KO mice, we totally understand the concerns of Reviewer #3 regarding a potential recirculation of activated T cells from the periphery. As noted by the Reviewer #3, though we analyzed neonatal animals (5-7 day old), we cannot exclude the possibility that the exacerbated frequency of Foxp3^{neg} Helios^{high} CD4^{pos}CD8^{low} TGFβR-KO cells could be due to thymic colonization from the periphery of T cells that acquire both CD4 and CD8 once in the thymus. In order to definitively rule out this hypothesis, we performed fetal thymus organ cultures (FTOC E17.5+ 5 days) as suggested by the Reviewer. These data, now shown in Supplementary figure 1 clearly show that the over-representation of Foxp3^{neg} Helios^{pos} cells in TGFβR-KO mice is not due to the side effects of peripheral inflammation or potential recirculation into the thymus of lymphocytes from the periphery. The text has been revised page 5-6 to underline this important conclusion obtained based on the Reviewer #3 suggestion.

Alternatively, could they be iNKT cells or IELp?

The possibility that the over-repressed FOXP3^{neg} Helios^{high} CD4^{pos}CD8^{low} cells are composed of either iNKT cells or IELp is really unlikely. Indeed, the thymus of TGFβR-KO mice was previously described to exhibit a large defect in both iNKT cell and IELp development (Li et al Immunity 2006, Doisnes et al J.Exp Med 2009, Kokel et al Nat Immunol 2011). However, in order to address the Reviewer concern, we performed a PBS-57 tetramer staining (specific to iNKT) as well as α4β7 and αE staining (specific to type A and B IELps). As illustrated below, the Foxp3^{neg} Helios^{high} CD4^{pos} CD8^{low} thymocytes from TGFβR-KO mice were not PBS-57 tetramer^{pos} thus not iNKT cells and they do not express high levels of α4β7 and αE associated with IELp.

Figure 1R3: iNKT and IELp analysis among Foxp3^{neg} Helios^{pos} CD4^{pos} CD8^{low} thymocytes from TGFβR-KO mice Flow cytometry staining iNKT (PBS-57 tet) on total thymocytes or Foxp3^{neg} Helios^{pos} CD4^{pos} CD8^{low} thymocytes from TGFβR-KO mice aged of 10 days. or Foxp3^{neg} Helios^{pos} CD4^{pos} CD8^{low} thymocytes PBS-57 tet^{neg} cells were then stained for α4β7 and CD103. On histograms, FMO is in blue and the indicated stainings are in red.

2) The link between impaired induction of AIRE⁺ mTEC, the aberrant T cell population and autoimmunity is a potentially important finding, but several aspects of this element of the study are puzzling.

Firstly, although RANKL levels are clearly reduced in the aberrant HELIOS^{pos} FOXP3^{neg} cells, the overall proportion of thymocytes expressing this ligand is comparable between WT and KO mice based on the dot plots in Figure 4A (proportions in the upper quadrants). Therefore, TGF β cannot be necessary for RANKL expression, nor can this mechanism be the complete explanation for why AIRE⁺ cells are not being induced. Are the CD4^{pos}CD8^{low} cells that are HELIOS^{high} FOXP3^{neg} producing something that interferes with RANK signaling in mTEC?

We totally agree with reviewer #3 that RANKL was expressed at similar levels on Foxp3^{neg} Helios^{neg} cells between TGF β -RWT and TGF β -RKO mice in figure 4A. This observation suggests that unlike Foxp3^{neg} Helios^{neg} thymocytes, the Foxp3^{neg} Helios^{pos} thymocytes require TGF- β signaling to express high levels of RANKL. This requirement of TGF- β signaling to highly autoreactive thymocytes was also observed to control CXCR3 expression (figure 3). Notably TGF- β signaling was reported to affect selectively other highly autoreactive thymocyte subsets including Tregs, or iNKT cells (Liu et al Nat Immunology 2007, Doines et al J.Exp Med 2009). Furthermore, RANKL has been shown as a key signal delivered by thymocytes to induce AIRE⁺ mTEC differentiation (Hikosaka Y et al Immunity 2008) and we previously demonstrated the key role of the signals delivered specifically by highly autoreactive thymocytes compared to weakly autoreactive cells in AIRE⁺ mTEC differentiation (Irla et al Immunity 2008).

Nevertheless, the control of RANKL production in Foxp3^{neg} Helios^{pos} thymocytes is likely not the unique mechanism by which TGF- β controls AIRE⁺ mTEC differentiation. Indeed, we also revealed here that TGF- β contributes to the medullary localization of highly autoreactive thymocytes and thus their unique ability to deliver signals (including RANKL) to mTEC responsible for their maturation. Thus, the impairment of AIRE⁺ mTEC differentiation in the absence of TGF- β is at least mediated by of two effects - namely the reduction levels of RANKL on highly autoreactive thymocytes and the impaired ability of these cells to reach the medulla and thus deliver signals to them. In order to avoid any confusion, this point has been highlighted in page 10.

Also, none of the autoimmune features shown in Figure 6 are normally observed in AIRE KO mice (I'm presuming here that all the mice used in this study are on an inbred C57BL/6 background – this is not stated in materials). Further development of this line of enquiry is essential to support this potentially interesting finding.

We apologize for only quoting the reference of our previous work describing the background of the mice we used in this manuscript. The mice were indeed on a C57BL/6 background and this point has been corrected in the method section.

The reviewer#3 is absolutely right, that on a C57BL/6 background, AIRE-KO mice show a mild autoimmune phenotype. Our present work reports that both medullary (AIRE-dependent) and cortical (AIRE-independent) negative selection are impaired in TGF β R-KO mice (cf. fig 2g). Of note, the numbers of cortical thymocytes that are negatively selected have been estimated to be 2–3 times higher than the numbers of negatively selected medullary thymocytes (Stritesky et al PNAS 2013 and Daley et al J.Exp. Med 2013). Thus, regarding the importance of cortical negative selection, the massive autoimmunity observed in TGF β R-KO mice, is likely a consequence of the defect in both cortical and medullary negative selection, which explain the difference in autoimmune features with AIRE-KO mice.

Moreover, in the BM chimera TGF β -RKO:WT mice, in which AIRE expression and medullary negative selection was restored, the mice developed autoimmune inflammation (Li et al Immunity 2006, Marie et al Immunity 2006), suggesting a larger contribution of the defect of the cortical negative selection in the autoimmunity syndromes we observed in TGF β -RKO mice. This point has been discussed in the revised version of the manuscript on page 14.

MINOR POINTS:

SuppFig 2; There is no measure of medullary volume. Please add quantification of this region to support the conclusion that the size of the medulla is unaffected in the TGF β RKO. It actually looks like there might be a more subtle impact, with many more small islets of medulla apparent in the KO compared to the WT. Is the localization of SP thymocytes normal?

We appreciate the suggestion to quantify medulla area in the TGF β RKO. This quantification is now been added in supplementary fig 3, and the related methods as well as the figure legend have been updated. No significant defect was observed in medullary area.

In order to remove any concern, we provide below to the Reviewer, another set of thymus sections from the same thymus presented in supplementary figure 2 of the previous version of the manuscript.

Figure 2 R3: Histology of the thymus

Immunostaining on thymus sections from TGF β R-KO mice and their littermate control (TGF β R-WT). The medulla was stained with the mTEC marker Keratin 14 (red) and counterstained with DAPI (blue).

As requested by the Reviewer#3, we also analyzed the localization of SP thymocytes. The positioning of CD4 and CD8 SP thymocytes into the medulla seemed normal in these mice, which is in accordance with the normal development of the thymic medulla (cf. Figure 2 R3 and supplementary figure 3a, b) beside the defect of AIRE. We are happy to provide a staining illustration of this result to the reviewer #3.

Figure 3R3

Thymic sections from TGF β R-KO mice and littermate controls were stained with antibodies against CD4 (green), CD8 (blue) and the mTEC specific marker Keratin 14 (K14, red);

There is a very large fraction of active caspase-3+ TEC, which is highly unlikely to represent the true rate of apoptosis in this population in vivo. Most of this cell death is likely to have been induced by the sample preparation. I suggest leaving these data out because they are not an adequate test of the hypothesis and not essential to the key finding that AIRE+ cells are reduced.

Based on reviewer #3's suggestion, these data have been removed.

Figure 3. The cortico-medullary junction (CMJ) is being defined as a 100um region extending from the K14+ mTEC boundary. Firstly, the zones indicated in Figure 3A seem rather variable (i.e. do not look like a uniform 100um from the K14 boundary). More detail of how the analysis of the CMJ were performed using ImageJ is required. Secondly, given that the authors have already established a defect in the maturation of AIRE+ mTEC (many of which reside near the CMJ), is it suitable to base this distinction using a mTEC marker like K14? A more suitable definition of the CMJ would be based on a cortical epithelial marker (e.g. beta 5t or K8), which may be normal in the TGFbRKO mice.

We appreciate the suggestion of reviewer #3 to better define the CMJ region to reinforce the message of Foxp3^{neg} Helios^{pos} cell localization in this thymic region. In order to avoid any concern of how the region was drawn, the analysis was reperformed by a recognized expert in image analysis using computational approaches (Dr A Sergé). The CMJ borders have been modified (cf. new Figure 3a), and the methods explained in detail in this revised version. Importantly, as the reviewer can appreciate the automated analysis has confirmed that the density of Foxp3^{neg} Helios^{pos} cells is higher in the CMJ of TGFbRKO mice than in their littermate controls (cf. new Figure 3b). Thanks to this unquestioned approach our data are now stronger.

Nevertheless, we are confused by the reviewer #3's concern of using K14 as the marker of the medulla border in our study. The K14 marker has been largely used by many groups to define the thymic medulla. Furthermore, if we established a defect in the expression of AIRE in mTEC (Figure 2) we did not observe a defect in mTEC. Indeed, as illustrated in supplementary Figure 3 neither the size of the medulla (we quantified based on the reviewer suggestion) nor the number of mTEC were affected TGFbR-KO mice. Importantly, AIRE-KO mice were described to not have a defect in medulla size (Anderson et al Science 2002). Moreover CXCL10, ligand of CXCR3 is enriched at the CMJ (Drennan J. Immunol 2009), in line with the CXCR3 helios^{pos} cell positioning at the CMJ.

The conclusion that the aberrant CD4^{pos}CD8^{low} HELIOS^{high} cells are prevented from export requires further support. Maturing thymocytes are able to leave the thymus without going to the medulla (e.g. CCR7KO mice). Could it be that these are activated T cells that have come into the thymus, and localized at the CMJ due to their Th1-like phenotype and expression of CXCR3?

The hypothesis formulated by Reviewer #3 can be indeed supported by the Th1-like phenotype of the Helios+ thymocytes in TGFbR-KO mice. We performed the FTOC experiment suggested by the reviewer (please see above). Since in the FTOC conditions, we observed an exacerbated proportion of CD4^{pos}CD8^{low} Helios^{high} cells (cf. new Supplementary Figure 1) that cannot be explained by a thymic colonization from the periphery, we believe that we can rule out this hypothesis.

Do the CD4^{pos}CD8^{low} cells that are HELIOS^{high} FOXP3^{neg} and the CD4^{SP} HELIOS^{high} cells express CCR7? Hard to tell from the histograms of the bulk population shown in 1B, and might be better shown by gating directly on these populations to assess CCR7 expression. This will provide important data on potential location. Likewise, what is the expression of CXCR4 (which is critical for the cortical localization of DP) and S1P1 (required emigration; this is shown for CD4^{SP} in Fig 5e but not his population).

As the reviewer requested, in order to avoid any confusion, we show CCR7 expression at the surface of both Helios^{pos} populations. As illustrated in supplementary Figure 5a, CCR7 expression was largely restricted to the more mature thymocytes (CD4^{SP}) in line with the ability of these cells to enter the medulla.

We happy to the Reviewer #3 provide the stainings requested for CXCR4 and S1P1. Since CD4^{pos}CD8^{low} cells correspond to a mature stage of DPs, only a fraction of them still expresses CXCR4. As expected the mature thymocytes (CD4^{SP}) failed to express this chemokine receptor (figure 4 R3). These observations are line with the FTOC experiments demonstrating that the over representation of Foxp3^{neg}Helios^{pos} CD4^{pos}CD8^{low} cells is not explained by the recirculation in the thymus from periphery, a phenomena which involves CXCR4. In contrats to their mature CD4 SP counterparts only a small fraction of Foxp3^{neg} Helios^{pos} CD4^{pos}CD8^{low} expressed S1P1 and this expression is independently of TGF-β signaling (figure 4 R3)

Figure 4R3 Expression of CXCR4 and S1P1

a) Flow cytometry of CXCR4 surface expression in Foxp3^{neg}Helios^{pos} CD4^{pos}CD8^{low} from TGFβR-KO mice and littermate controls. DP thymocytes are used as positive control. **b)** Flow cytometry S1P1 surface expression in Foxp3^{neg}Helios^{pos}. CD4 Foxp3^{neg}Helios^{pos} CD4 SP thymocytes are used as positive control according figure 5e.

Figure 5f. The recent thymic emigrant experiment shown here is a good one, but the authors restrict their analysis to only HELIOS expression among the RTE. They should show a broader comparison of the overall FITC+ RTE in WT vs KO. Are there more RTE in the KO? What is their predominant phenotype? These data might provide more insight into thymic function and the preferential export of cells between the genotypes.

We appreciate reviewer #3 comments on experiment figure 5f. As suggested by the Reviewer, to improve the message, figure 5f now shows the percentage of total CD3^{pos} FITC^{pos} cells which was similar between the two types of mice suggesting a similar thymus output.

Figure 6. Some quantification of the autoimmune phenotypes observed is essential. Did all mice show these infiltrates? How severe were they? For instance, the images of pancreas shown demonstrate some insulinitis, but there does not appear to be destruction of the islet.

We thank the reviewer for the suggestion to provide some quantification of the pathology observed. Additional data have been added to figure 6.

All animals were sacrificed before they reach limiting points fixed by our local ethics committee (hunching, diarrhea, for more than 2 days in a row). Diabetes is first preceded by peri-islet infiltration, then destruction. Clinical signs are usually observed when more than 80% of the islet are destroyed. Likely, the animals, which show colitis and dermatitis were sacrificed before the islets were destroyed. In order to avoid any concern, we completed the text on page 12 by providing the notion of pre-diabetes associated with islet infiltration without sign of destruction.

Reviewers' comments:

Reviewer #1 (Remarks to the Author):

McCarron et al have satisfactorily addressed the concerns raised in my previous review of their manuscript outlining a key role for TGFb in promoting the negative selection of developing thymocytes. AS mentioned previously this work will be of interest to the numerous researchers interested in TGFb effects on T cell biology and is important to publish. The authors have included new data to address my previous concerns and where this was not possible they have appropriately discussed why this could not occur. The manuscript is ready for publication.

Reviewer #3 (Remarks to the Author):

The revised version provides new data and discussion that support the authors conclusions. All of the minor points I raised were well addressed. However, some concerns about the major points raised remain, outlined below:

1. Could the aberrant population of CD4+CD8low HELIOShigh FOXP3neg cells represent recirculating activated T cells?

I appreciate that the authors points and they have now added an additional experiment investigating the aberrant cells arising in fetal thymic organ culture (FTOC); however, only one FACS profile showing a modest effect is shown. Was this reproducible? The legend states it is a representative profile, but no information regarding how many replicates and experiments were performed is provided. Graphs quantifying this population in WT and KO FTOC should be shown, along with statistical analysis (as is done well through the rest of the ms) to support the interpretation that these cells arise independent of inflammation.

2. Why don't the other RANKL-positive cells induce AIRE expression?

The data showing that HELIOS-high thymocytes have impaired RANKL is clear. Yet, the authors also show that there are normal numbers of HELIOS-low mature thymocytes in the thymus expressing RANKL and that these reside in the medulla. My query is about why these cells cannot induce normal AIRE expression, and whether an alternative mechanism might explain the deficiency. The authors contend that previous literature establishes that it is only RANKL expression by the HELIOS-high cells (which are presumably autoreactive) that is relevant for AIRE induction. However, I am not convinced that the HELIOS-high cells are the exclusive mediators of AIRE expression.

The authors quote the (very nice) 2008 Irla et al, Immunity paper; however, these data involved either MHC II-deficient mTECs or TCR tg systems to conclude that high avidity interaction promoted AIRE+ mTEC. They did not establish that low avidity interactions with RANKL-pos cells in a polyclonal setting (like the mice studied here) cannot induce substantial AIRE expression. Furthermore, other cell types expressing RANKL have been shown to induce AIRE expression without being "autoreactive" (e.g. lymph node inducer cells, gd Tcells, (Roberts et al., 2012, Immunity). I wonder whether something else is missing and/or that RANKL signals are being antagonized (e.g. via OPG production).

Regardless, I believe that deeper discussion of this issue is warranted, highlighting further evidence to support the author's interpretation of this key point and/or acknowledgement of uncertainties about why AIRE induction is so poor.

3. If AIRE-dependent negative selection is impaired, why is the autoimmunity so severe?

The author's interpretation that cortical deletion is impaired rests entirely on the notion that HELIOS is

an unequivocal measure of autoreactivity, which is debatable. Other models have reported increases in HELIOS⁺ cortical thymocytes of similar or greater magnitude, yet do not exhibit the autoimmune phenotypes described here (e.g. the Bim^{-/-} or Vav-BCL-2tg mice in the Daley et al. JEM study). The authors should consider other interpretations, such as changes in thymocyte/T cells stimulation thresholds, as a consequence of impaired TGFβ signaling.

Editorial requests:

1. Please make sure the data presentation in the manuscript matches all relevant the requirements listed in the Reporting Checklist. For example, flow cytometry plots should be presented either as pseudocolor or contour plots. Plain dot plots like Fig1a do not reflect data point density and are not allowed per journal reporting standards.
2. Please consider including a cartoon summarizing the new findings of this work, to be included as a main or supplementary figure.
3. Please provide Source data as one Excel file, where data for each figure are provided in separate sheets.

Reviewer #3

The revised version provides new data and discussion that support the authors conclusions. All of the minor points I raised were well addressed. However, some concerns about the major points raised remain, outlined below:

We appreciated that Reviewer #3 underlined that our revised version provided new data and discussion that support our conclusions

1. Could the aberrant population of CD4+CD8low HELIOShigh FOXP3neg cells represent recirculating activated T cells?

I appreciate that the authors points and they have now added an additional experiment investigating the aberrant cells arising in fetal thymic organ culture (FTOC); however, only one FACS profile showing a modest effect is shown. Was this reproducible? The legend states it is a representative profile, but no information regarding how many replicates and experiments were performed is provided. Graphs quantifying this population in WT and KO FTOC should be shown, along with statistical analysis (as is done well through the rest of the ms) to support the interpretation that these cells arise independent of inflammation.

We are sorry for not providing statistical analysis on supl figure 1. The 2.5 fold change was validated on several FTOCs. Of note, this approach is an in vitro development which not fully recapitulates the physiological conditions, but that was powerful enough to rule out the retro-migration concern. Once again, we thank the reviewer for suggesting the FTOC experimental approach.

2. Why don't the other RANKL-positive cells induce AIRE expression?

The data showing that HELIOS-high thymocytes have impaired RANKL is clear. Yet, the authors also show that there are normal numbers of HELIOS-low mature thymocytes in the thymus expressing RANKL and that these reside in the medulla. My query is about why these cells cannot induce normal AIRE expression, and whether an alternative mechanism might explain the deficiency. The authors contend that previous literature establishes that it is only RANKL expression by the HELIOS-high cells (which are presumably autoreactive) that is relevant for AIRE induction. However, I am not convinced that the HELIOS-high cells are the exclusive mediators of AIRE expression.

The authors quote the (very nice) 2008 Irla et al, Immunity paper; however, these data involved either MHC II-deficient mTECs or TCR tg systems to conclude that high avidity interaction promoted AIRE+ mTEC. They did not establish that low avidity interactions with RANKL-pos cells in a polyclonal setting (like the mice studied here) cannot induce substantial AIRE expression. Furthermore, other cell types expressing RANKL have been shown to induce AIRE expression without being "autoreactive" (e.g. lymph node inducer cells, gd Tcells, (Roberts et al., 2012, Immunity). I wonder whether something else is missing and/or that RANKL signals are being antagonized (e.g. via OPG production).

Regardless, I believe that deeper discussion of this issue is warranted, highlighting further evidence to support the author's interpretation of this key point and/or acknowledgement of uncertainties about why AIRE induction is so poor.

We are confused by Reviewer#3 comment since we believe that we never claimed that "it is only RANKL expression by the HELIOS-high cells that is relevant for AIRE induction". However, if the Reviewer raised this point is that we were not clear and we apology for that. As suggested by the Reviewer, we revised the discussion by clearly underlying that the lack of AIRE is not only due to RANKL defect on Helios cells. We reminded that defect on the ability of Helios cells to reach the medulla and thus the impairment of interplay mTEC /autoreactive cells (which could include RANK signal but also CD40 signal and other unknow). We also proposed potential impairment of signal provided by other thymocytes without mentioning the LTI since the role of this later was only reported during embryogenesis before E16. Regarding the hypothesis on OPG and RANKL antagonist, we did not develop this point since AIRE expression is only partially impaired in the TGFβR-KO mice.

3. If AIRE-dependent negative selection is impaired, why is the autoimmunity so severe?

The author's interpretation that cortical deletion is impaired rests entirely on the notion that HELIOS is an unequivocal measure of autoreactivity, which is debatable. Other models have reported increases in HELIOS+ cortical thymocytes of similar or greater magnitude, yet do not exhibit the autoimmune

phenotypes described here (e.g. the Bim^{-/-} or Vav-BCL-2tg mice in the Daley et al. JEM study). The authors should consider other interpretations, such as changes in thymocyte/T cells stimulation thresholds, as a consequence of impaired TGF β signaling.

We appreciate the reviewer #3 suggestion and the hypothesis on the TCR activation threshold has been added in the discussion in addition to figure 7 which gives an overview of the different effects of TGF- β signaling in thymocytes on four key steps important for negative selection (including cortical negative selection but not only).

REVIEWERS' COMMENTS:

Reviewer #3 (Remarks to the Author):

The authors have fully responded to my queries and I recommend publication.